# Sample weighting as an explanation for mode collapse in generative adversarial networks

## Abstract

Generative adversarial networks were introduced with a logistic **MiniM**ax cost formulation, which normally fails to train due to saturation, and a **N**on-**S**aturating reformulation. While addressing the saturation problem, NS-GAN also inverts the generator's sample weighting, implicitly shifting emphasis from higher-scoring to lower-scoring samples when updating parameters. We present both theory and empirical results suggesting that this makes NS-GAN prone to mode dropping. We design MM-nsat, which preserves MM-GAN sample weighting while avoiding saturation by rescaling the MM-GAN minibatch gradient such that its magnitude approximates NS-GAN's gradient magnitude. MM-nsat has qualitatively different training dynamics, and on MNIST and CIFAR-10 it is stronger in terms of mode coverage, stability and FID. While the empirical results for MM-nsat are promising and favorable also in comparison with the LS-GAN and Hinge-GAN formulations, our main contribution is to show how and why NS-GAN's sample weighting causes mode dropping and training collapse.

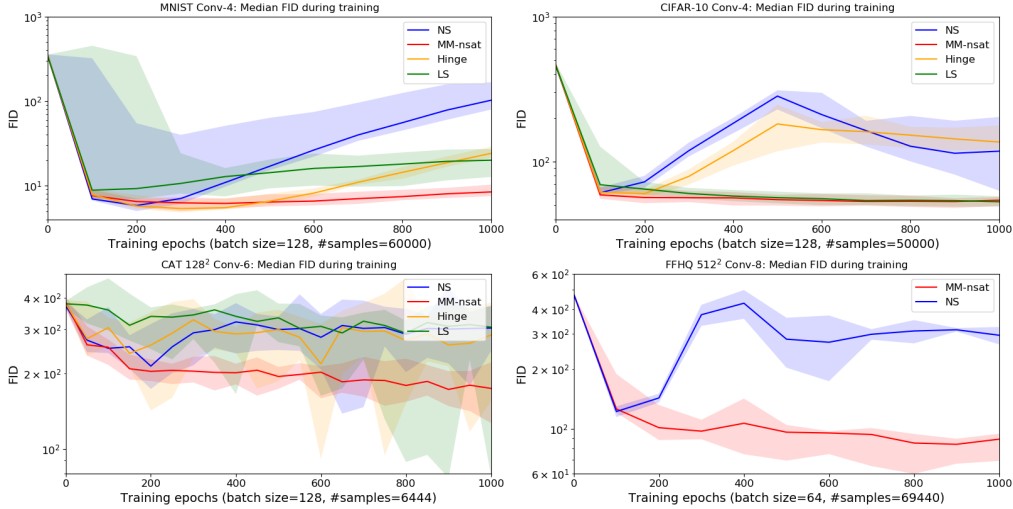

Figure 1: Median Fréchet Inception Distance during training for ten runs on MNIST, CIFAR-10, CAT $128^2$ and FFHQ $512^2$, using very simple convolutional GANs. The shaded areas show minimum and maximum value during training for the cost formulations. MM-nsat is best overall, suffers less from gradual mode dropping and trains reliably on the more challenging datasets.

# 1 Introduction

Generative adversarial networks have come a long way since their introduction (Goodfellow et al., 2014) and are currently state of the art for some tasks, such as generating images. A combination of deep learning developments, GAN specific advances and vast improvements in data sets and computational resources have enabled GANs to generate high resolution images that require some effort to distinguish from real photos (Zhang et al., 2018; Brock et al., 2018; Karras et al., 2018).

GANs use two competing networks: a generator $G$ that maps input noise to samples mimicking real data, and a discriminator $D$ that outputs estimated probabilities of samples being real rather than generated by $G$. We summarize their cost functions, $J_D$ and $J_G$, for the minimax and non-saturating formulations introduced in Goodfellow et al. (2014). We denote samples from real data and noise distributions by $x$ and $z$ and omit the proper expectation value formalism:

$$J_{D_{MM}}(x,z) = J_{D_{NS}}(x,z) = -\log(D_p(x)) - \log(1 - D_p(G(z)))$$
$$J_{G_{MM}}(z) = \log(1 - D_p(G(z))) \tag{1}$$
$$J_{G_{NS}}(z) = -\log(D_p(G(z))$$

For clarity, we use subscripts to distinguish between the discriminator's pre-activation logit output $D_l$ and the probability representation $D_p$:

$$D_p \equiv (1 + \exp(-D_l))^{-1} \tag{2}$$

Both formulations have the same cost function for $D$, representing the cross entropy between probability estimates and ground truth. In the minimax formulation (MM-GAN), $G$ is simply trained to maximize $D$'s cost. Ideally, $G$ matches its outputs to the real data distribution while also achieving meaningful generalization, but many failure modes are observed in practice. NS-GAN uses a modified cost for $G$ that is non-saturating when $D$ distinguishes real and generated data with very high confidence, such that $G$'s gradients do not vanish. (Supplementary: C)

Various publications establish what the different cost functions optimize in terms of the Jensen-Shannon and reverse Kullback-Leibler divergences between real and generated data:

$$J_{G_{MM}} \Leftrightarrow 2 \cdot D_{JS} \qquad \text{(Goodfellow et al., 2014)}$$
$$J_{G_{MM}} + J_{G_{NS}} \Leftrightarrow D_{KL}^R \qquad \text{(Huszár, 2016)} \tag{3}$$
$$J_{G_{NS}} \Leftrightarrow D_{KL}^R - 2 \cdot D_{JS} \quad \text{(Arjovsky \& Bottou, 2017)}$$

Huszár (2015) and Arjovsky & Bottou (2017) have suggested NS-GAN's divergence as an explanation for the ubiquitous mode dropping and mode collapsing problems with GANs (Metz et al., 2016; Salimans et al., 2016; Srivastava et al., 2017). While MM-GAN seems promising in terms of its Jensen-Shannon divergence, the formulation has largely been ignored because the saturating cost causes training to break down.

A variety of other GAN formulations have been introduced, such as WGAN-GP (Arjovsky et al., 2017; Gulrajani et al., 2017), LS-GAN (Mao et al., 2016) and Hinge-GAN (Miyato et al., 2018). Lucic et al. (2018) finds that different cost formulations tend to get similar results given sufficient parameter tuning, including various forms of regularization. Despite the questionable form of NS-GAN in terms of divergences, it is widely used and can produce very impressive results, such as in the improved StyleGAN (Karras et al., 2019).

## 2 THEORY

### 2.1 MM-GAN SATURATION

The parameters of a network are typically trained with some form of gradient descent on a cost function. We find the expressions for $D$'s and $G$'s gradients with respect to their parameters, $\phi$ and $\theta$: (Supplementary: F)

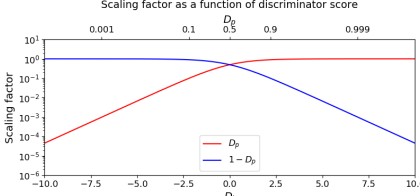

$$\nabla_\phi J_{D_{MM,NS}} = +\frac{\partial D_l(G(z), \phi)}{\partial \phi} \cdot D_p(G(z), \phi)$$

$$+ \frac{\partial D_l(x, \phi)}{\partial \phi} \cdot (1 - D_p(x, \phi))$$

$$\nabla_\theta J_{G_{MM}} = -\frac{\partial D_l(G(z), \theta)}{\partial \theta} \cdot D_p(G(z), \theta) \tag{4}$$

$$\nabla_\theta J_{G_{NS}} = -\frac{\partial D_l(G(z), \theta)}{\partial \theta} \cdot (1 - D_p(G(z), \theta)))$$

Figure 2: Scaling factors as a function of the discriminator output, at a scale emphasizing asymptotic behaviors. MM-GAN's scaling factor causes $G$'s gradient to vanish when $D_p(G(z)) \to 0$, which corresponds to the maximum value of its cost.

We emphasize the two kinds of *scaling factors* for these gradients in red and blue: they are plotted in figure 2. The discriminator's scaling factors decrease as it minimizes its cost, approaching 0 towards the optima for both the real and the generated data term.

The minimax formulation $J_G = -J_D$ is suitable for adversarial training in terms of the generator's optimum, but the unchanged scaling factor means that $G$'s gradients increase towards and decrease away from its optimum. The saturation effect described in Goodfellow et al. (2014) is that $\lim_{D_p(G(z)) \to 0} \nabla_\theta J_{G_{MM}} = 0$, such that $G$ stops training when $D$ is highly confident that its samples are fake. More generally, the scaling factor makes $J_G$ concave with respect to $D_l$, which interacts poorly with common optimization methods (see section 2.4).

As $\nabla_\theta J_{G_{MM}}$ and $\nabla_\theta J_{G_{NS}}$ are the same aside from their scaling factors, the different behaviors of the two formulations must follow from these. NS-GAN's scaling factor avoids saturation, but gives rise to a different, more subtle mode dropping tendency (see section 2.3).

## 2.2  NON-SATURATION AND SAMPLE WEIGHTING

As can be seen from eq 4, the NS-GAN and MM-GAN gradients are parallel for a *single sample*, but with different magnitudes. Stochastic gradient descent estimates the gradient of the cost over the entire input distribution by using a number of samples (a minibatch). We can express the NS-GAN minibatch gradient in terms of the MM-GAN gradient:

$$\nabla_\theta J_{G_{NS}}^{\text{batch}} = \sum_{i=0}^{N-1} \nabla_\theta J_{G_{MM}}^{\text{sample}}(z_i) \left[ \frac{1 - D_p(G(z_i))}{D_p(G(z_i))} \right] \tag{5}$$

Due to the bracketed factor, NS-GAN rescales the contribution from each sample relative to MM-GAN, implicitly emphasizing samples with smaller values of $D_p$. Seeing as saturation is caused by the gradient's vanishing magnitude, this additional effect on the gradient's direction is questionable.

The exact ratio of the minibatch gradient magnitudes for NS-GAN and MM-GAN depends on $\partial/\partial\theta(D_l(G(z)))$ for each sample and has no convenient expression. We can approximate it by replacing $D_p(G(z_i))$ in eq 5 with its mean over the minibatch, $\overline{D_p} = \frac{1}{N} \sum_{i=0}^{N-1} D_p(G(z_i))$. This allows us to formulate a form of non-saturation for MM-GAN that mimicks NS-GAN:

$$\nabla_\theta J_{G_{MM\text{-}nsat}}^{\text{batch}} = \frac{1 - \overline{D_p}}{\overline{D_p}} \sum_{i=0}^{N-1} \nabla_\theta J_{G_{MM}}^{\text{sample}}(z_i) \tag{6}$$

We refer to the formulation with this generator gradient as **MM-nsat**. The relative weights of samples in each batch are as for MM-GAN, while the gradient magnitude approximates that of NS-GAN. Note, however, that the relative weights of samples may be disturbed across batches, such as when the minibatch size is small and $\overline{D_p}$ fluctuates. Despite different theoretical motivation, MM-nsat is very closely related to importance weighted NS-GAN (Hu et al., 2017). (Supplementary: D)

## 2.3  SAMPLE WEIGHTING AND MODE DROPPING

If we use $r(x)$ and $g(x)$ to denote the density of real and generated samples at a point $x$ in data space, the optimal discriminator (i.e. the function that minimizes $J_{D_{MM,NS}}$) is given by:

$$D_p^{\text{opt}}(x) = \frac{r(x)}{r(x) + g(x)} \text{(Goodfellow et al., 2014)} \tag{7}$$

This expression for the optimial discriminator assumes idealized conditions: that $D$ is optimized for a fixed $G$, with unlimited capacity and without having to estimate the underlying real and generated data distributions by finite sampling. While $D^{\text{opt}}$ is not realized in practice (Sinn & Rawat, 2018), we use it to form some intuitions about $D$'s behaviors.

Suppose we have real data with two disjunct, equiprobable modes, and that one of these modes, $\mathbb{O}$, is overrepresented in generated data. For convergence, $G$ would need to shift probability mass from $\mathbb{O}$ to the underrepresented mode, $\mathbb{U}$. However, the minibatches used to update $G$ will have more samples from $\mathbb{O}$, simply because they are generated more often. For a strong discriminator, these samples will also tend towards smaller values of $D_p(G(z))$, due to equation 7. This in turn

causes them to be weighted differently by NS-GAN and MM-GAN, because $D_p(G(z))$ appears in their scaling factors (eq 4). We refer to the first effect as over- and underrepresentation (of generated samples relative to real samples) and the second effect as up- and downweighting (governed on the scaling factor).

The fundamental problem with NS-GAN can be seen by looking at how these two effects interact. The NS-GAN generator has a scaling factor $1 - D_p(G(z))$, which combines overrepresentation with upweighting and underrepresentation with downweighting, allowing an overrpresented mode $\mathbb{O}$ to dominate $\mathbb{U}$'s contributions to the parameter updates. If parameter updates overwhelmingly based on gradients from $\mathbb{O}$ have an adverse effect on $G$'s ability to generate samples from $\mathbb{U}$, this mode may become increasingly underrepresented, making $\mathbb{O}$ yet more dominant.

MM-GAN's scaling factor has the reverse behavior, pairing overrepresentation with downweighting and underrepresentation with upweighting, such that the two effects combine in a stabilizing rather than destabilizing way. For the extreme example of a nearly dropped or newly discovered mode, $r(x) \gg g(x)$ such that we expect $D_p(x) \approx 1$ for a strong discriminator. NS-GAN's sample weighting disregards gradients from such samples, whereas MM-GAN's sample weighting emphasizes them.

This difference between MM-GAN and NS-GAN in terms of scaling factors reflects the different divergences they have been shown to optimize (eq 3). Huszár (2015) and Arjovsky et al. (2017) both connect the mode dropping tendencies of NS-GAN to its reverse Kullback-Leibler divergence, which strongly penalizes $G$ for generating samples outside of the real data modes, but not for dropping modes. A variety of attempts to address mode dropping, mode collapse and mode hopping in GANs seem unaware that this is reasonable behavior for NS-GAN given its divergence: mode dropping minimizes $D_{KL}^R$ and maximizes $D_{JS}$.

## 2.4 MM-GAN INTERACTION WITH ADAM

GANs are generally trained with the Adam optimizer (Kingma & Ba, 2014), as was recommended by Radford et al. (2015) while introducing the DCGAN architecture. The parameter update step for Adam is given by:

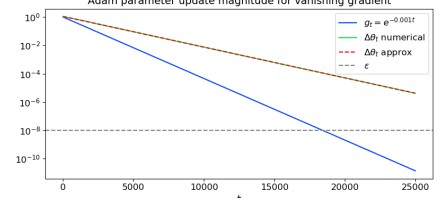

$$\Delta\theta_t = -\alpha \frac{\hat{m}_t}{\sqrt{\hat{v}_t} + \epsilon}$$

$$m_t = \beta_1 m_{t-1} + (1 - \beta_1)g_t, \quad \hat{m}_t = \frac{m_t}{1 - \beta_1^t} \qquad (8)$$

$$v_t = \beta_2 v_{t-1} + (1 - \beta_2)g_t^2, \quad \hat{v}_t = \frac{v_t}{1 - \beta_2^t}$$

Here, $g_t$ is the gradient at timestep $t$, and $\hat{m}_t$ and $\hat{v}_t$ are the first and second order exponential moving averages of the parameterwise gradients, bias-corrected to account for zero-initialization. There are four hyperparamters: $\epsilon$ is a small constant primarily for numerical stability, $\alpha$ is

Figure 3: Update step size for an artificial gradient $g_t = \exp(-0.001t)$ with $\beta_1 = 0.99$, $\beta_2 = 0.999$ and $\alpha = 1$. In green and red, simulated and approximated update magnitudes, using equations 8 and 11. Updates vanish even while $g_t \gg \epsilon$.

the learning rate and $\beta_1$ and $\beta_2$ determine the effective *memories* of the moving averages.

The fraction $\frac{\hat{m}_t}{\sqrt{\hat{v}_t}}$ resembles unit normalization of a vector, and for a constant gradient $g_t = g$ (such that the moving averages are trivial) update steps depend only on the sign of $g$, if $|g| \gg \epsilon$.

$$\Delta\theta_t = -\alpha \frac{\text{sgn}(g)}{1 + \frac{\epsilon}{|g|}} \qquad (9)$$

However, this normalization does not necessarily address MM-GAN's saturation problem, due to the training dynamics. $D_l^{\text{opt}} = \pm\infty$ where the real and generated data do not overlap (eq 7), such that if $D$ can cleanly separate real from generated samples, we expect it to further decrease its loss by inflating its outputs. Supposing that $D$ approaches this optimum linearly, i.e. $D_l^t(G(z)) = at$, we get:

$$D_p^t(G(z)) \approx \exp(at) \qquad (10)$$

$D_p$ also appears as the MM-GAN scaling factor (eq 4), such that $G$ will be optimized with gradients of the form $g_t = g_0 \exp(at)$. For reasonable values of $a$, $\beta_1$ and $\beta_2$, the update step size for each of

$G$'s parameters can be approximated: (Supplementary: G)

$$\Delta\theta_t \approx g_0 C \exp((a - \frac{1}{2}\log(\beta_2))t) \tag{11}$$

Given the commonly used $\beta_2 = 0.999$, parameter updates will vanish exponentially if $a < -0.0005$, given that $|D_l|$ increases fast enough and conforms reasonably well to our simplified model. If $D$ learns to distinguish the real and generated data manifolds before they meaningfully intersect, this interaction between $D$, $G$ and Adam threatens to freeze parameter updates altogether. (Supplementary: L)

## 3 METHOD

### 3.1 COST FUNCTION MODIFICATIONS

In addition to our novel non-saturating version of MM-GAN (MM-nsat: eq 6), we normalize the gradient magnitudes of the original cost functions (MM-unit, NS-unit). Motivated by our model for MM-GAN saturation in section 2.4, we also test if modifying the Adam $\beta_2$ parameter for $G$ gives the expected results (MM $\beta_2 = 0.99$). (Supplementary: H)

MM-nsat is the only of these cost functions that is interesting in its own right: the others are used to demonstrate various behaviors and highlight the roles of sample weighting as opposed to gradient magnitude. Note that the prefixes MM and NS always correspond to the sample weighting used by the cost function. Furthermore, gradient magnitudes are matched for MM-unit & NS-unit and approximately matched for MM-nsat & NS-GAN.

### 3.2 EVALUATION

Evaluating a generator is generally difficult. In addition to visual inspection, we use Frechét Inception Distance (Heusel et al., 2017), which we find informative also on MNIST, in spite of the feature extraction network being trained on natural images. Due to our focus on the mode dropping problem, we also use an unusual metric for datasets with class labels: we compute the **Jensen-Shannon divergence between class distributions** in real and generated data. For $G$, we estimate its class distribution by drawing samples and using a pre-trained classification network to label them. Note that this metric only sees class distributions and is not sensitivite to the mode coverage or collapse *within* any given class. (Supplementary: I)

$$D_{JS}CD \equiv D_{JS}(\text{class frequencies}_{\text{dataset}}||\text{class frequencies}_{\text{generator}}) \tag{12}$$

### 3.3 EXPERIMENTS

For the majority of our experiments we use very simple networks without tuning their hyperparameters: either fully connected networks (FC) with a fixed number of hidden units, or strided convolutional networks with kernel size 3 and doubling the number of filters when the width and height is halved (Conv). We use ReLU activations, except for the final layer, where $G$ uses tanh activation with real data normalized to $[-1, 1]$, while $D$ uses sigmoid activation to map $D_l$ to $D_p$.

We make use of how fully connected networks are harder to train than convolutional ones (Thanh-Tung et al., 2018) and how training grows increasingly fragile for deeper networks and higher resolution datasets to find illustrative test cases. We make use of batch normalization (Ioffe & Szegedy, 2015) (bn), zero centred real data gradient penalty (Mescheder, 2018) (sgp) and spectral normalization (Miyato et al., 2018) (sn) only where explicitly mentioned. In addition to the simpler networks, we test the full DCGAN (Radford et al., 2015) and StyleGAN (Karras et al., 2018) architectures.

We primarily use the MNIST (LeCun & Cortes, 2010) and CIFAR-10 (Krizhevsky et al., 2009) datasets to investigate the differences between cost functions. We run additional experiments using the CAT (Zhang et al., 2008; Jolicoeur-Martineau, 2018) and FFHQ (Karras et al., 2018) datasets to study behaviors for higher resolution images.

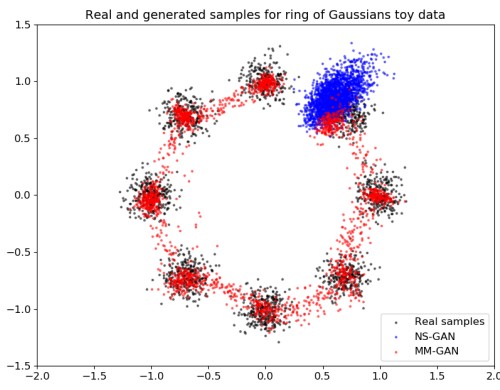

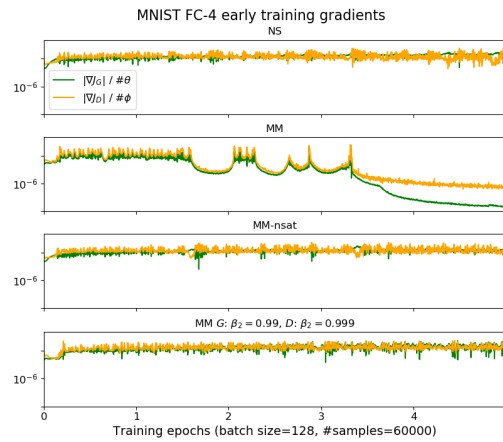

Figure 4: Real and generated samples on a classic toy 2D mixture of Gaussians dataset: a *sample* is here a coordinate pair. In blue, NS-GAN samples exhibiting mode dropping behavior, here in the process of *hopping* between modes. In red, samples using the unmodified MM-GAN cost function. MM-GAN trains well on this toy problem despite its saturating cost function and shows reasonable coverage of all modes.

Figure 5: Gradient magnitudes in the early phase of training on MNIST with 4-layer fully connected networks. Top two: NS-GAN working properly and unmodified MM-GAN in its saturating failure mode, ultimately freezing parameters. Bottom two: MM-nsat, and MM-GAN with $G$'s Adam parameter reduced $\beta_2 = 0.999 \rightarrow 0.99$, both avoiding saturation.

## 4 RESULTS AND DISCUSSION

### 4.1 QUALITATIVE PRELIMINARIES

Training GANs on a ring of Gaussians has been used to study both the mode dropping tendency of GANs (Metz et al., 2016; Srivastava et al., 2017) and how to address it. MM-GAN tends to do very well on such problems, as shown in figure 4. For this problem, it is easy to generate samples indistinguishable from real ones, limiting problems with saturation. MM-GAN is more mode-covering in practice as suggested by its divergence (Huszár, 2015; Arjovsky et al., 2017). (Supplementary: S)

Using MNIST and weak, fully-connected networks, we replicate the well-known failure mode of MM-GAN that motivates the NS-GAN reformulation. In figure 5, we show the gradient magnitudes early in training, in particular a super-exponentially vanishing gradient for MM-GAN after roughly 3 epochs that halts training altogether. Additionally, we show that reducing $\beta_2$ only for $G$'s optimizer (effectively giving it a shorter memory for the second order momentum) stabilizes the training process, as suggested by our theory on the interaction of MM-GAN and Adam. (Supplementary: L)

For the same setup, we show samples at the end of prolonged training for NS and MM-nsat cost functions in figure 6. We see that our version of minimax non-saturation trains well. The difference in terms of mode coverage is visually striking and corresponds well with our numerical evaluations for the same generators in table 1.

For MNIST, NS-GAN's mode collapse can mostly be addressed by early stopping or regularization. In figure 7 we show samples from training on the more challenging CAT $128^2$ dataset. We find that MM-nsat trains well and generates fairly realistic samples, whereas for NS-GAN, catastrophical mode collapse occurs before $G$ learns to produce reasonable samples. More rigorous and quantitiative results are presented in the supplementary material, particularly U and V.

### 4.2 QUANTITATIVE EVALUATION

Figure 8 shows more comprehensive results for MNIST and CIFAR-10. We plot metrics throughout training to emphasize the dynamics for MM-GAN and NS-GAN, as well as variants of the same cost functions. There are two key results: that simple stabilization techniques allow us to train GANs with

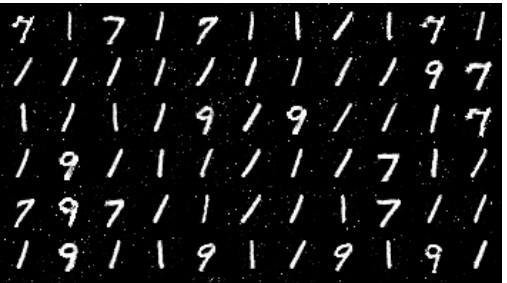

NS-GAN: FID $= 152$, $D_{JS}CD = 0.56$

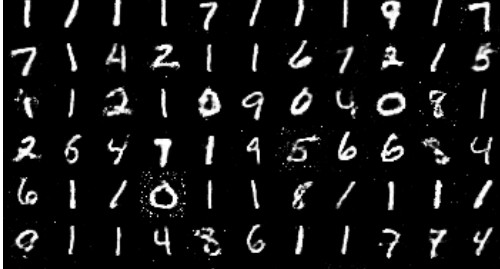

MM-nsat: FID $= 49.7$, $D_{JS}CD = 0.15$

Figure 6: Generated samples after 1000 epochs for MNIST with 4-layer fully connected networks. MM-nsat is strikingly better in terms of mode coverage.

Table 1: Class distributions for $G$ (see fig 6).

| Cost | Class frequency in % | | | | | | | | | |
|---|---|---|---|---|---|---|---|---|---|---|
| | 0 | 1 | 2 | 3 | 4 | 5 | 6 | 7 | 8 | 9 |
| NS | 0 | 75 | 0 | 0 | 0 | 0 | 0 | 15 | 0 | 10 |
| MM-nsat | 5 | 45 | 3 | 4 | 9 | 5 | 7 | 12 | 3 | 8 |

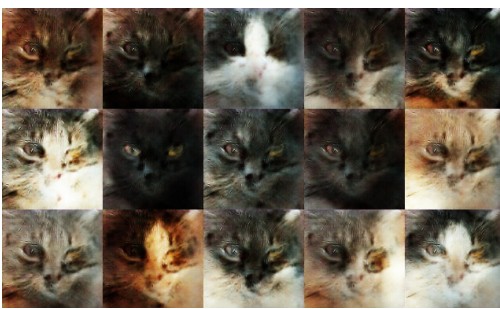

Early stopping NS samples (best run)

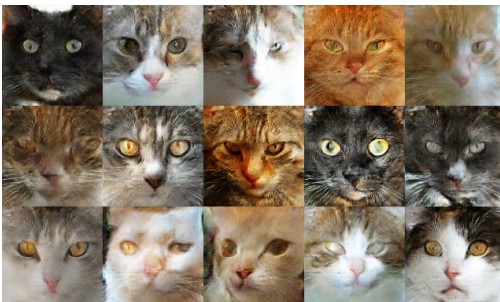

Early stopping MM-nsat samples (worst run)

Figure 7: Samples from training runs for CAT $128^2$ using the DCGAN architecture with spectral normalization in the discriminator and self-attention. All NS runs collapse in terms of diversity, compared to only one of the MM-nsat runs. We show the least favorable comparison for MM-nsat, i.e. the run with *latest* collapse for NS and earliest collapse for MM-nsat. For each of these runs, we hand pick the best looking batch of samples generated during training.

MM-GAN sample weighting, without the saturation issues that plague unmodified MM-GAN; and that their behavior is qualitatively different from those using NS-GAN sample weighting. Gradient magnitudes mostly influence training stability. Additionally, the plots on the right hand side show that the performance gain for minimax variants is reflected by more correct class distributions in generated data. In some cases, such as for 3-layer fully connected networks on MNIST, there is a notable gap between $D_{JS}CD$ values, even while FID is similar.

The importance of sample weighting is best demonstrated by comparing MM-unit and NS-unit. By construction, these variants have the same gradient magnitude, such that the only difference between the formulations is whether high- or low-scoring samples are emphasized when finding the total gradient for a minibatch of samples. The results show that this difference is crucial for the training dynamics.

Aside from stability, there are no clear differences between the variants with minimax sample weighting. While MM-nsat and MM-unit avoid saturation in very different ways, their results are highly similar, to each other and to the results for unmodified MM-GAN when it does not saturate. This suggests that the approximation we make in equation 6 when introducing $\overline{D_p}$ is of limited importance, and that MM-nsat is more faithful to the original, logistic minimax formulation than NS-GAN.

The main issue with NS-GAN for these datasets is its strong tendency to deteriorate in terms of both FID and $D_{JS}CD$ as training progresses. There is an early stopping point where NS is competitive with MM-nsat in some cases, but not all. The results for CIFAR-10 are the most extreme: NS and NS-unit consistently undergo catastrophical mode collapse.

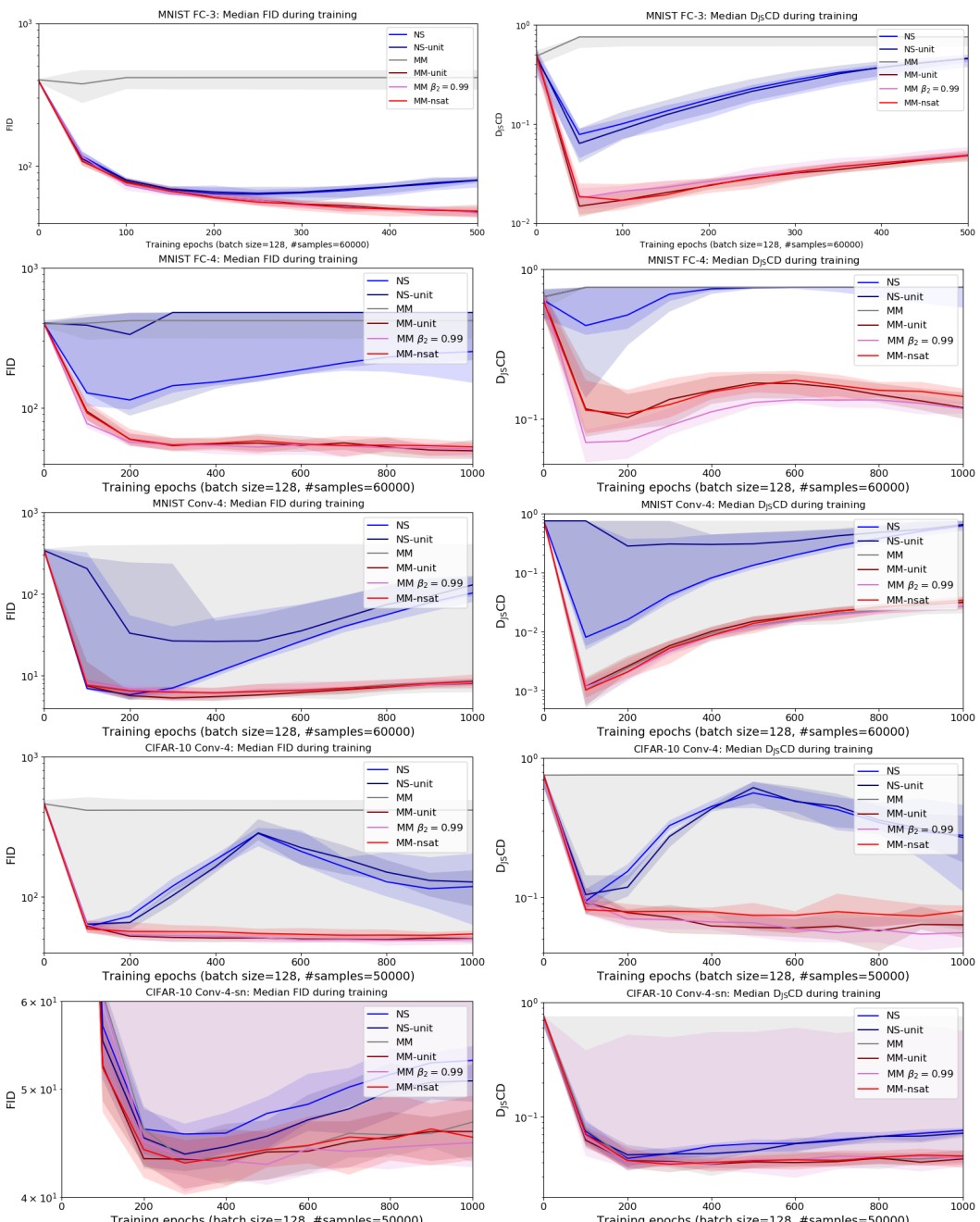

Figure 8: Left: FID, an estimated distance between generated and real data based on Inception activations. Right: $D_{JS}CD$, the Jensen-Shannon divergence between class distributions in generated and real data. For various networks using the MNIST and CIFAR-10 datasets, we plot the median values for ten runs during training. The shaded area indicates maximum and minimum values. NS and MM are the traditional cost functions; variants use normalized gradients, reduced $\beta_2$ for $G$'s Adam optimizer and the non-saturating rescaling from eq 6.

Overall, these results validate theory from the literature showing that MM-GAN and NS-GAN correspond to different divergences and the importance of sample weighting as described in section 2.3. We present a variety of additional results in the supplementary material (M through W), both testing the validity of our theory in different experimental settings and showing results for higher dimensional datasets.

## 5 CONCLUSION AND FURTHER WORK

Based on a more thorough, theoretical analysis of the differences between the formulations introduced in Goodfellow et al. (2014), we have designed a form of non-saturation for the minimax cost function for GANs that rescales the minibatch gradient. This corrects the training difficulties for the original minimax GAN, without the side-effects inherent to the NS-GAN reformulation. Running experiments, we have shown that our new stabilization has qualitatively different behavior, in particular better mode coverage as indicated both by our theory and by previous works showing which divergences are optimized by NS-GAN and MM-GAN. Perhaps most importantly, our work is an important correction to the view that NS-GAN is just MM-GAN corrected for its saturation problems, as the name might suggest.

We have shown promising results with MM-nsat (our gradient rescaled version of MM-GAN), but it is primarily designed for demonstration purposes. Results with simpler networks on MNIST and CIFAR-10 are much stronger for MM-nsat compared to NS-GAN, Hinge-GAN and LS-GAN, and experiments on higher resolution images from the CAT and FFHQ datasets show that MM-nsat has less issues with catastrophic mode collapse during the early stages of training, greatly reducing the stability issues that GANs tend to suffer from.

Interactions with discriminator regularization are unclear and important for more advanced applications (see R). Performance when combining MM-nsat with various designs that directly address mode dropping remains to be determined, for instance the class conditioning (Mirza & Osindero, 2014) used in BigGAN (Brock et al., 2018) and the minibatch standard deviation used in StyleGAN (Karras et al., 2018). Unlike the normalizations and regularizations that mask the underlying issues with NS-GAN and often introduce extra hyperpameters, MM-nsat changes only the cost function.

While we have focused on variants of MM-GAN and NS-GAN, with the convenience that these all work with the same discriminator cost function, we expect analysis in terms of sample weighting to apply much more generally, explaining training dynamics and tradeoffs between individual sample quality and overall sample diversity also for other GAN formulations.

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

Table 2: Overview of GAN cost formulations.

| Cost | Reference | *D* cost | *G* sample emphasis | *G* gradient magnitude |
|------|-----------|----------|---------------------|------------------------|
| MM-GAN | Goodfellow et al. (2014) | Cross-entropy | High-scoring | Saturating |
| MM-nsat | *ours* | Cross-entropy | High-scoring | $\approx$ Non-saturating |
| MM-unit | *ours* | Cross-entropy | High-scoring | Unit-normalized |
| MM $\beta_2 = 0.99$ | *ours* | Cross-entropy | High-scoring | Saturating, adj. Adam |
| NS-GAN | Goodfellow et al. (2014) | Cross-entropy | Low-scoring | Non-saturating |
| NS-unit | *ours* | Cross-entropy | Low-scoring | Unit-normalized |
| LS-GAN | Mao et al. (2016) | Quadratic | ? | - |
| Hinge-GAN | Miyato et al. (2018) | Clipped linear | Uniform | - |

## SUPPLEMENTARY MATERIAL

## A    TABLE OF COST FUNCTIONS

Refer to table 2 for an overview of the cost functions used in this paper. Note that sample emphasis is not directly comparable between formulations with different costs for the discriminator.

## B    INFORMAL AND EXTENDED SUMMARY

*We include an extended summary of the most central points from our work. This presentation is intended to be less technical and somewhat more accessible to a wider audience.*

When training a classifier, the standard cross entropy loss has stronger gradients for samples with greater error. This means that correcting misclassifications is emphasized over further increasing the confidence level of correctly classified samples. Training usually runs for multiple epochs, giving the classifier repeated opportunities to readjust if its cost increases for any particular sample.

Like for classifiers, every term in the NS-GAN cost functions for the discriminator and generator emphasizes the samples which are furthest from their optima. While this seems reasonable, it does not necessarily interact well with adversarial training.

To minimize its cost when training on equal amounts of real and generated data, the discriminator $D_p(x)$ approximates $P(y \text{ is real}|y = x)$. This means that $D_p \approx 1$ for real data that $G$ does not generate and $D_p \approx 0$ for generated data outside of the real data manifold. More generally, $D_p(x)$ approximates the density of real samples at $x$ relative to the sum of densities of real and generated samples.

The generator cost function includes a real data term which is often omitted: without any gradient with respect to $G$'s parameters it does not influence the training. This makes it difficult for $G$ to act on dropped modes, because its updates only see $D$'s gradients at the points in data space where $G$ generates samples.

The degree to which NS-GAN's generator cost emphasizes samples further from their optimum is quantified by the scaling factor given in equation 4. NS-GAN weights contributions from each generated sample by a factor $1 - D_p$, such that low-scoring ("unrealistic") samples contribute the most to the gradient.

MM-GAN's generator instead uses the scaling factor $D_p$, emphasizing high-scoring ("realistic") samples. MM-GAN saturation follows directly from this scaling factor: for highly unrealistic generated samples, $D_p \rightarrow 0$, and this weighting causes $G$'s gradient vanish. Adam does not work around this problem with the usual hyperparameter settings.

In either case, $G$'s parameters are updated by sampling the gradient of $D$ with respect to $G$'s parameters at points $x$ where samples happen to be generated. This means that the updates see most contributions from regions where the density of generated samples is high.

The main problem with NS-GAN is how this sampling frequency interacts with its scaling factor. Regions on the real data manifold where the density of generated samples is too low tend towards small values of $1 - D_p$: these generated samples are considered realistic by $D$ and thus have small

error for $G$ and are weighted down by the scaling factor. Since this region in data space is also sampled rarely, due to the $G$'s low density, its contribution to $G$'s parameter updates will be small. At the same time, regions with too high density of generated samples have large scaling factors and are sampled frequently. In a situation where different modes disagree on the best configuration of $G$'s parameters, undersampled modes will be massively outvoted.

This would not be a problem if $D$'s gradients were able to push the density of generated samples away from oversampled modes and discover undersampled or entirely dropped modes. However, we have no guarantee that this sort of *action at a distance* will work in practice: the literature instead shows that mode collapse is a recurring problem, likely because $D$'s gradients fail to be informative across longer distances in data space.

Sample weighting also explains why MM-GAN behaves much better in terms of mode coverage. Since MM-GAN uses the opposite scaling factor, $D_p$, it boosts the contributions to parameter updates from undersampled modes. The MM-GAN scaling factor partially cancels against the sampling bias effect, making the parameter update less dependent on a single, dominant mode. While a generated sample in an undersampled mode is not itself an *error*, the very high value of $D_p$ for this generated sample reflects a large discrepancy between generated and real densities at this point. MM-GAN's sample weighting makes it better able to address errors of this sort.

The main problem with MM-GAN is that its gradient decreases when the cost is increases. This makes it very difficult to optimize, also with Adam. NS-GAN, the established solution to this problem, changes both sample weighting and gradient magnitudes. The problem can instead be addressed with a very simple rescaling of the minibatch gradient, as shown in eq 10. We call the resulting formulation **MM-nsat** as it combines minimax sample weighting with the same form of non-saturation as used in Goodfellow's NS-GAN. The key difference is that the NS-GAN non-saturation is implicitly applied to each sample, disturbing sample weights, while the MM-nsat rescaling is explicitly applied to the minibatch gradient as a whole.

Our experimental results validate that this minibatch gradient rescaling corrects the saturation problem, and comparing NS-GAN and MM-nsat allows us to show that sample weighting is highly important for training dynamics. Mode coverage and general stability is greatly improved for MM-nsat relative to NS-GAN, which sometimes suffers from gradual mode dropping throughout training, and sometimes from catastrophical mode collapse early in training. Note, however, that strong models already tend to include various designs to counteract the problems caused by NS-GAN's unfortunate sample weighting, limiting the benifits of simply replacing NS-GAN with MM-nsat.

## C   GOODFELLOW'S MOTIVATION OF NS-GAN

We include Goodfellow et al. (2014)'s motivation for NS-GAN for easy reference: "In practice, equation 1 [minimax cost] may not provide sufficient gradient for $G$ to learn well. Early in learning, when $G$ is poor, $D$ can reject samples with high confidence because they are clearly different from the training data. In this case, $\log(1 - D(G(z)))$ saturates. Rather than training $G$ to minimize $\log(1 - D(G(z)))$ we can train $G$ to maximize $\log D(G(z))$. This objective function results in the same fixed point of the dynamics of $G$ and $D$ but provides much stronger gradients early in learning."

## D   IMPORTANCE WEIGHTING

Hu et al. (2017) studies the parallels between generative adversarial networks and variational autoencoders. They transfer the idea of importance weighting from VAE to GANs, arriving at an update rule for the generator (Hu et al. (2017): equation 22) which reweights $G$'s NS-GAN gradients for each sample in the minibatch with a normalized weighting factor. Closely related ideas are seen in Hjelm et al. (2018) and Che et al. (2017). The expression for the unnormalized weighting factor is given as:

$$w_i = \frac{q_{\phi_0}^r(y|\mathbf{x}_i)}{q_{\phi_0}(y|\mathbf{x}_i)} \tag{13}$$

Here, $q_{\phi_0}$ and $q_{\phi_0}^r$ represent the discriminator and its reverse respectively, i.e. $q_{\phi_0}^r(y|\mathbf{x}) = q_{\phi_0}(1 - y|\mathbf{x})$, where $y$ represents the true label of a sample $\mathbf{x}$. In our notation, this is corresponds to the

following, as given explicitly in Hjelm et al. (2018):

$$w_i = \frac{D_p(G(z_i))}{1 - D_p(G(z_i))} \tag{14}$$

Comparing this weighting factor to the bracketed factor in our eq 5, it is clear that it perfectly cancels against the reweighting of NS-GAN relative to MM-GAN. In other words, importance weighted NS-GAN actually recovers the same original MM-GAN sample weighting that is used by MM-nsat, such that our approach of avoiding the NS-GAN sample reweighting leads us to rediscover this update rule.

The remaining difference is that of the gradient magnitude. When reweighting the NS-GAN gradients, Hu et al. (2017) and Hjelm et al. (2018) choose to normalize the weights over the minibatch: omitting to do this would make their importance weighting also recover the MM-GAN gradient magnitude, and thus reintroduce the MM-GAN saturation problem. They normalize by calculating the actual sum of the reweighting factors, which is slightly different from our approximation where we replace $D_p$ with its mean over the minibatch (eq 6). Note that neither method is exact unless the samplewise gradients in the minibatch have the same magnitude. As seen by for instance comparing MM-nsat and MM-unit, which have highly similar performance despite massively different gradient magnitudes, these diferences should have little effect on training dynamics.

The primary difference between importance weighthing for NS-GAN and our design of MM-nsat is the logical steps and the theoretical motivation. In our work, the direct relationship to the original MM-GAN update rule is made explicit, whereas importance weighting goes the cirucuitous route of first taking the heuristic NS-GAN formulation in place the original MM-GAN formulation, then reweighting it to arrive at an importance weighted NS-GAN formulation, which our work shows to actually be just a non-saturating version of MM-GAN.

## E    CONCURRENT WORK

A concurrent work recently made available as a pre-print, Sinha et al. (2020), makes use of a simple design which is directly comparable to our method. When updating $G$, they ignore the gradients arising from the lowest scoring generated samples, where the fraction of the samples ignored increases throughout training. This effectively shifts weighting from low-scoring to high-scoring samples, which is the same overall effect as is achieved by using our MM-nsat gradient. Their method is much simpler and less principled as to exactly how low- and high-scoring samples are weighted, but it is also more easily applied to other GAN cost formulations.

Similar to our findings, Sinha et al. (2020) observe an increase in mode coverage and overall quality. By investigating the cosine similarity between low- and high-scoring samples, they observe that optimization conflicts between these, as we suggest in section 2.3. Their analysis in terms of pushing samples towards and away from modes is somewhat at odds with ours, where we note that low-scoring samples can be high-quality, but lie in modes that are significantly oversampled by $G$. The discriminator score is not a direct measure of the sample's quality, but rather of relative densities of real and generated data, such that we should not expect $D$'s gradients to necessarily point towards the centres of the real data modes.

## F    GRADIENTS DERIVED FROM COST FUNCTIONS

Finding the MM-GAN gradient for a single sample is a matter of straightforward calculus. The expression for NS-GAN's gradient can be found using the same approach. First note that the relationship between $D_p$ and $D_l$ is given by eq 2, such that:

$$\frac{\partial D_p}{\partial D_l} = D_p^2 \exp(-D_l) \tag{15}$$

$$\exp(-D_l) = \frac{1 - D_p}{D_p} \tag{16}$$

We take $J_G = -J_D$ (eq 1) as the starting point, as defines the minimax formulation. Note that the first term does not depend on $G$ or $\theta$, such that it can be omitted in the $G$ cost function:

$$J_{G_{\text{MM}}} = \log(D_p(x)) + \log(1 - D_p(G(z,\theta)))$$

$$
\begin{aligned}
\nabla_\theta J_{G_{\text{MM}}} &= -\frac{\partial D_p(G(z,\theta))}{\partial \theta} \cdot \frac{1}{1 - D_p(G(z,\theta))} \\
&= -\frac{\partial D_l(G(z,\theta))}{\partial \theta} \cdot \frac{\partial D_p}{\partial D_l} \cdot \frac{1}{1 - D_p(G(z,\theta))} \\
&= -\frac{\partial D_l(G(z,\theta))}{\partial \theta} \cdot \frac{D_p^2 \exp(-D_l)}{1 - D_p} \\
&= -\frac{\partial D_l(G(z,\theta))}{\partial \theta} \cdot \frac{D_p(1 - D_p)}{(1 - D_p)} \\
&= -\frac{\partial D_l(G(z,\theta))}{\partial \theta} \cdot D_p(G(z,\theta))
\end{aligned}
\tag{17}
$$

It might seem counterintuitive to mix $D_l$ and $D_p$ when expressing the gradients, instead of using for instance the expression on line 2 of equation 17. However, the non-linear behavior of $\partial D_p/\partial \theta$ makes interpretation difficult. As $D_p$ approaches either 0 or 1, increasingly small changes in $D_p$ reflect the same change in odds ratios. It is less clear from the gradient in terms of only $D_p$ that the MM-GAN gradient saturates as $D_p$ approaches 0, or that it does not diverge as $D_p$ approaches 1. However, for the scaling factor itself, we find the $D_p$ formulation most intuitive. The interpretation suggested by eq 7 is particularly convenient.

## G  MM-GAN INTERACTION WITH ADAM

We include additional details for the approximation of the Adam update step. We combine equation 8 and $g_t = g_0 \exp(at)$. For simplicity, we assume that $\text{sgn}(g_t)$ does not depend on $t$: if the sign of the gradient changes, $m_t$ is reduced and $v_t$ unaffected. This can contribute to making $\Delta\theta_t$ vanish, but cannot prevent it.

This allows us to express the momentum as a sum of contributions from each previous time step $i$.

$$m_t = \sum_{i=1}^{i=t} \beta_1^{t-i}(1 - \beta_1)g_0 \exp(ai) \tag{18}$$

For reasonably large values of $t$ and assuming $a \neq \log(\beta_1)$, we can approximate this sum by a definite integral:

$$
\begin{aligned}
m_t &\approx \int_{i=0}^{i=t} (1 - \beta_1)\beta_1^{t-i} g_0 \exp(ai) di \\
&= \frac{1 - \beta_1}{a - \log(\beta_1)} g_0(\exp(at) - \beta_1^t)
\end{aligned}
\tag{19}
$$

Using this approximation (similarly also for $v_t$), assuming $\epsilon$ to be negligible and packing assorted time independent constants including $-\alpha$ into $C$, we get the update step:

$$\Delta\theta_t \approx g_0 C \frac{\sqrt{1 - \beta_2^t}}{1 - \beta_1^t} \frac{\exp(at) - \exp(\log(\beta_1)t)}{\sqrt{|\exp(2at) - \exp(\log(\beta_2)t)|}} \tag{20}$$

Having already assumed $t$ to be large, we omit the first fraction (bias corrections) for simplicity:

$$\Delta\theta_t \approx g_0 C \frac{\exp(at) - \exp(\log(\beta_1)t)}{\sqrt{|\exp(2at) - \exp(\log(\beta_2)t)|}} \tag{21}$$

For sufficiently large values of $t$, behavior is determined by which of the terms in the numerator and denominator have the largest constant in the exponent (i.e. slowest decay). We enumerate all the

four possible cases:

$$
\begin{aligned}
a > \log(\beta_1) \wedge a > \frac{1}{2}\log(\beta_2) &\rightarrow \Delta\theta_t \propto 1 \\
a < \log(\beta_1) \wedge a < \frac{1}{2}\log(\beta_2) &\rightarrow \Delta\theta_t \propto (\frac{\beta_1}{\sqrt{\beta_2}})^t \\
a > \log(\beta_1) \wedge a < \frac{1}{2}\log(\beta_2) &\rightarrow \Delta\theta_t \propto e^{(a-\frac{1}{2}\log(\beta_2))t} \\
a < \log(\beta_1) \wedge a > \frac{1}{2}\log(\beta_2) &\rightarrow \Delta\theta_t \propto e^{(\log(\beta_1)-a)t}
\end{aligned}
\tag{22}
$$

Using hyperparameters such that $\beta_1 > \sqrt{\beta_2}$ is highly unusual. It is required to satisfy the inequalities for the last case and means that the base is greater than one in the second case. In both cases, we get exploding update steps, which is generally undesirable.

Assuming $\beta_1 < \sqrt{\beta_2}$, the first three cases show intended behavior: update steps are normalized for constant or even exploding gradients, while they diminish for vanishing gradients, enabling convergence. The third case, $a < \frac{1}{2}\log(\beta_2)$, is discussed in the main text. In the second case, where $a < \log(\beta_1)$, we get a different kind of vanishing behavior governed by the relationship between $\beta_1$ and $\beta_2$.

Additional results and discussion for this problem is given in L.

## H    IMPLEMENTATION DETAILS

For consistency with the theoretical framework and to allow testing of magnitude normalized gradients, we modify GANs by rescaling the generator gradient. This amounts to calculating the minibatch gradient explicitly and multiplying it by a factor $R$ calculated as given in algorithms 1 and 2 before passing it on to the optimizer. (Code repository: J)

---
**Algorithm 1** Non-saturating minimax rescaling factor (MM-nsat)

---
**Require:** $G, D_l$: generator and logit valued discriminator functions
**Require:** $N$: the minibatch size
**Require:** $z_0, \ldots, z_{N-1}$: $N$ generator inputs
**Require:** $\epsilon_R$: a small constant, prevents zero-division overflow such that $R_{\max} = \epsilon_R^{-1}$
  1: $\overline{D_p} = \frac{1}{N}\sum_{i=0}^{i=N-1}(1 + \exp{(-D_l(G(z_i)))})^{-1}$
  2: $R \leftarrow \frac{1-\overline{D_p}}{\epsilon_R + \overline{D_p}}$

---

---
**Algorithm 2** Unit rescaling factor (MM-unit, NS-unit)

---
**Require:** $\nabla_\theta$: the gradient of the cost function with respect to $G$'s parameters
**Require:** $N_\theta$: the number of parameters of $G$
**Require:** $\epsilon_R$: a small constant, prevents zero-division overflow such that $R_{\max} = N_\theta\epsilon_R^{-1}$
  1: $|\nabla_\theta| = (\sum_{i=0}^{i=N_\theta-1}\nabla_{\theta_i}^2)^{\frac{1}{2}}$
  2: $R \leftarrow \frac{N_\theta}{\epsilon_R + |\nabla_\theta|}$

---

However, for proper behavior with penalty terms as well as convenience, we strongly recommend implementing MM-nsat by multiplying the minimax cost by a factor $R$ which depends on $D(G(z))$ for the entire batch and disabling back-propagation through this scaling factor. In TensorFlow, a straightforward implementation is:

$$
\texttt{cost\_mm-nsat = tf.stop\_gradient(R) * cost\_mm} \tag{23}
$$

The rescaling factor is constant across the minibatch and can be applied either samplewise or to the total cost.

For MM-unit and NS-unit, the rescaling factors unfortunately cannot be calculated without first computing the gradient for the unmodified cost functions.

Applying gradient rescaling directly does not interact well with penalty terms. When the MM-GAN cost saturates, the gradient will typically be dominated by the penalties, causing both MM-nsat and MM-unit rescaling to inflate their effect. Using the approach above, rescale the adversarial minimax cost and leave penalty terms unchanged.

Parallelization may raise some issues when calculating the rescaling factor $R$, because proper minimax behavior requires $R$ to be calculated across all parallels and a naive approach will introduce some overhead. Ideally, $D(G(z))$ and gradients from parallels should be aggregated in the same step, in order to rescale gradients before running them through the optimizer. Correctly implemented, the overhead should be unnoticeable for multi-GPU training.

We note that for very small batch sizes, or where $R$ for other reasons vary significantly between subsequent batches, MM-nsat will have some NS-like behaviors, because weighting between samples in different batches is disturbed by the difference between their values of $R$. However, not even unmodified MM-GAN weights properly across batches, due to updates to $D$ and optimizer corrections. It is possible to smooth $R$ across batches using for instance an exponential running mean, at the risk of reintroducing saturation issues.

In algorithm 2, we make the unintuitive choice of normalizing the gradient not to unity, but to $N_\theta$, the number of network parameters. Due to cancellation effects in Adam's parameter update step, this time-independent scaling factor is only relevant when $g_t$ is small and $\epsilon$ non-negligible. Because Adam's $\Delta\theta_t$ is calculated for each parameter independently, this choice of normalization prevents introducing an inadvertent relationship between the number of parameters of a network and whether updates vanish due to $\epsilon$ as shown in eq 9.

## I    CLASS DISTRIBUTION DIVERGENCE

Implementing a class distribution divergence requires us to have labels for the real data and some way to determine which class generated samples belong to. This limits its applicability to simpler problems where we can train strong classifiers. An additional complication is that we are applying this classifier to generated data, where it may not generalize well. The classifier is calibrated for real data samples and its behavior is not well defined outside of this manifold. In particular, a standard classifier is compelled by design to assign a class to all samples, even when they are so poor that they cannot meaningfully be said to belong to any class at all. It is conceivable that a generator producing only noise-like samples will be assigned functionally random classes and achieve a very good $D_{JS}CD$. Furthermore, $D_{JS}CD$ is not sensitive to the degree of coverage or collapse *within* any given class.

In practice, we find that a poor generator invariably scores badly in terms of $D_{JS}CD$, with nearly all its samples being assigned to the same class, and that FID and $D_{JS}CD$ are strongly correlated. Nonetheless, combining $D_{JS}CD$ with a standard generator evaluation method such as FID is recommended, and for poor generators, $D_{JS}CD$ may be misleading. Similarly, the class divergence serves as a control for our use of FID for MNIST, which is significantly removed from the natural images FID's Inception network in trained on.

For interpretation, note that we use a normalized Jensen-Shannon divergence, such that $D_{JS} \in [0, 1]$. The lower bound corresponds to perfectly matched distributions and increasingly mode dropping generators will approach the upper bound. When the real data distribution is non-zero for all classes, $(D_{JS}CD)_{max} < 1$. For the case where there are ten evenly distributed classes in real data, which includes MNIST and CIFAR-10, we get $(D_{JS}CD)_{max} \approx 0.758$.

For classification, we train a simple, convolutional network for MNIST and a ResNet (He et al., 2015) based convolution model with batch normalization (Ioffe & Szegedy, 2015) for CIFAR-10 with approximately 99% and 89% validation accuracy (see section J). For MNIST-1K, we apply the MNIST classifier to each channel separately, treating each ordered triplet of classifications as its own class. In line with the established standard for FID, we use 50000 samples for estimating $D_{JS}CD$ for a generator.

## J    CODE AND NETWORKS

We have made code for Python 2.7 and TensorFlow available in a GitHub-repository: to maintain anonymity, see uploaded supplementary material instead. Qualitatively reproducing the MNIST and toy experiments in the main paper is straightforward, but other datasets as well as running the evaluation metrics involves additional setup (see repository for details).

The repository includes code for training classifiers for $D_{JS}CD$ for MNIST and CIFAR-10. Step by step setup for the CAT and FFHQ datasets is **not** included, nor is the forked version of StyleGAN for training it with MM-nsat.

## K    GRADIENT NORM AND DIRECTION FOR MM-NSAT

Figure 9 shows how the gradients obtained by the NS-GAN and MM-nsat generator costs differ during training, both in terms of norm and direction. These results illustrate the theory from section 2.2 and give an impression of the quality of the approximation used for MM-nsat (eq 6), in order to renormalize its gradient magnitude of MM-GAN to match with that of NS-GAN.

For 4-layer convolutional networks, FID improves for both formulations with very similar quantitative results for roughly 200 epochs. During this period, gradients are also the most similar in terms of norm and direction. Afterwards, cosine similarity drops, indicating that gradients are becoming increasingly different for the purpose of optimization, and NS-GAN and MM-nsat diverge in terms of FID due to NS-GAN increasingly dropping modes.

For 4-layer fully connected networks, cosine similarity drops much earlier, and FID for NS-GAN falls behind very early in training, reflefcting mode dropping (see D$_{JS}$CD for FC-4 in figure 8). During the later stages of training, gradient norms diverge, particularly for the case where we train with the NS-GAN gradient. Note, however, that this is a stage of training where NS-GAN shows very pathological behavior, with more than $\frac{3}{4}$ of generated samples representing the digit 1.

For interpretation of the cosine similarity, note that gradients are high dimensional vectors, such that near-orthogonality is unexceptional. Gradient magnitudes are not perfectly matched for NS-GAN and MM-nsat, particularly not later in training where values of $D_p$ become more extreme and NS-GAN starts falling off in terms of FID. When training with the MM-nsat gradient, its norm relative to the NS-GAN gradient are matched to within an order of magnitude and allows MM-nsat to train without any apparent issues.

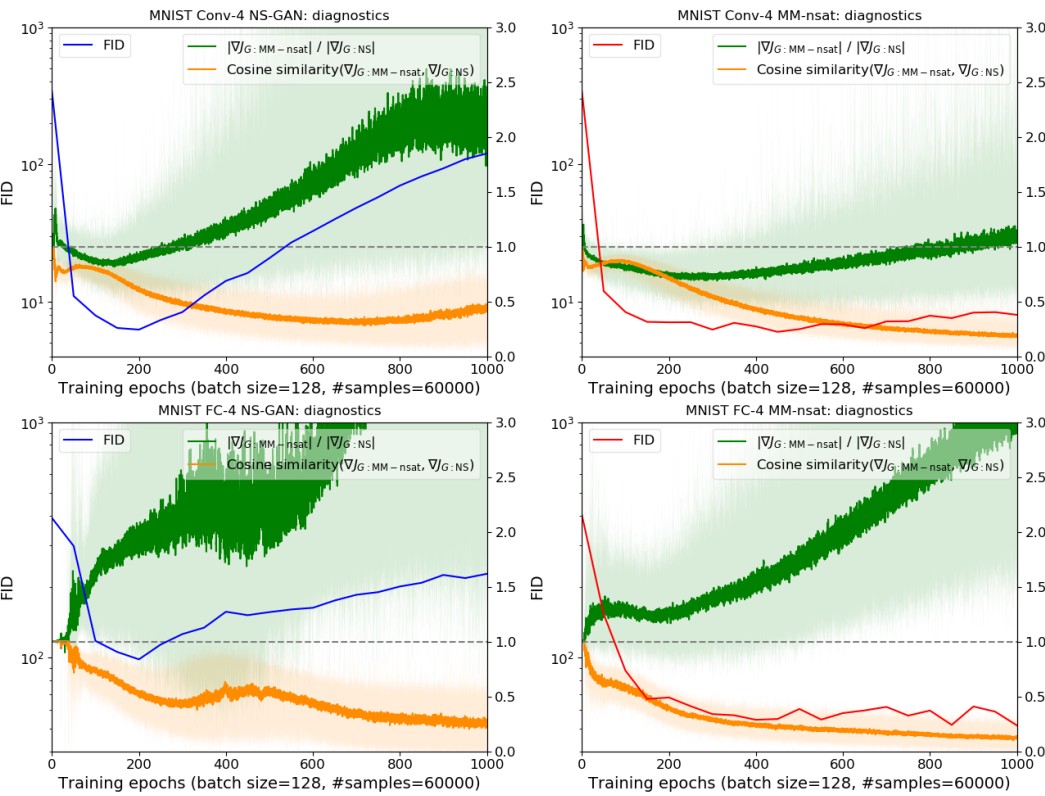

Figure 9: FID during training with additional diagnostics for 4-layer convolutional (top) and fully-connected (bottom) networks on MNIST, using (left) NS-GAN and (right) MM-nsat for the parameter updates. For each run, we calculate both the NS-GAN and MM-nsat gradients at each iteration and plot their relative norms (green) and cosine similarity (orange). We plot the running means, with the actual, noisy values indicated by the shaded areas.

Figure 10 shows the same kind of plots as figure 9, but comparing the original MM-GAN with MM-nsat instead. As the gradients obtained by these formulations are parallel by construction, the cosine similarity is almost exactly 1.0 at all times. We use differently scaled y-axes for the relative gradient magnitudes, as these approach $10^{16}$ when MM-GAN falls into its saturating failure mode.

The primary difference in gradient magnitude between MM-GAN and MM-nsat is a relatively brief spike very early in training, corresponding well with the the observation by Goodfellow et al. (2014) cited in section C. This spike is both taller and wider in the FC-4 setting, likely reflecting $G$ having more difficulty learning to generate somewhat realistic samples. During the rest of the training, the magnitude of MM-nsat relative to MM-GAN increases slowly and consistently.

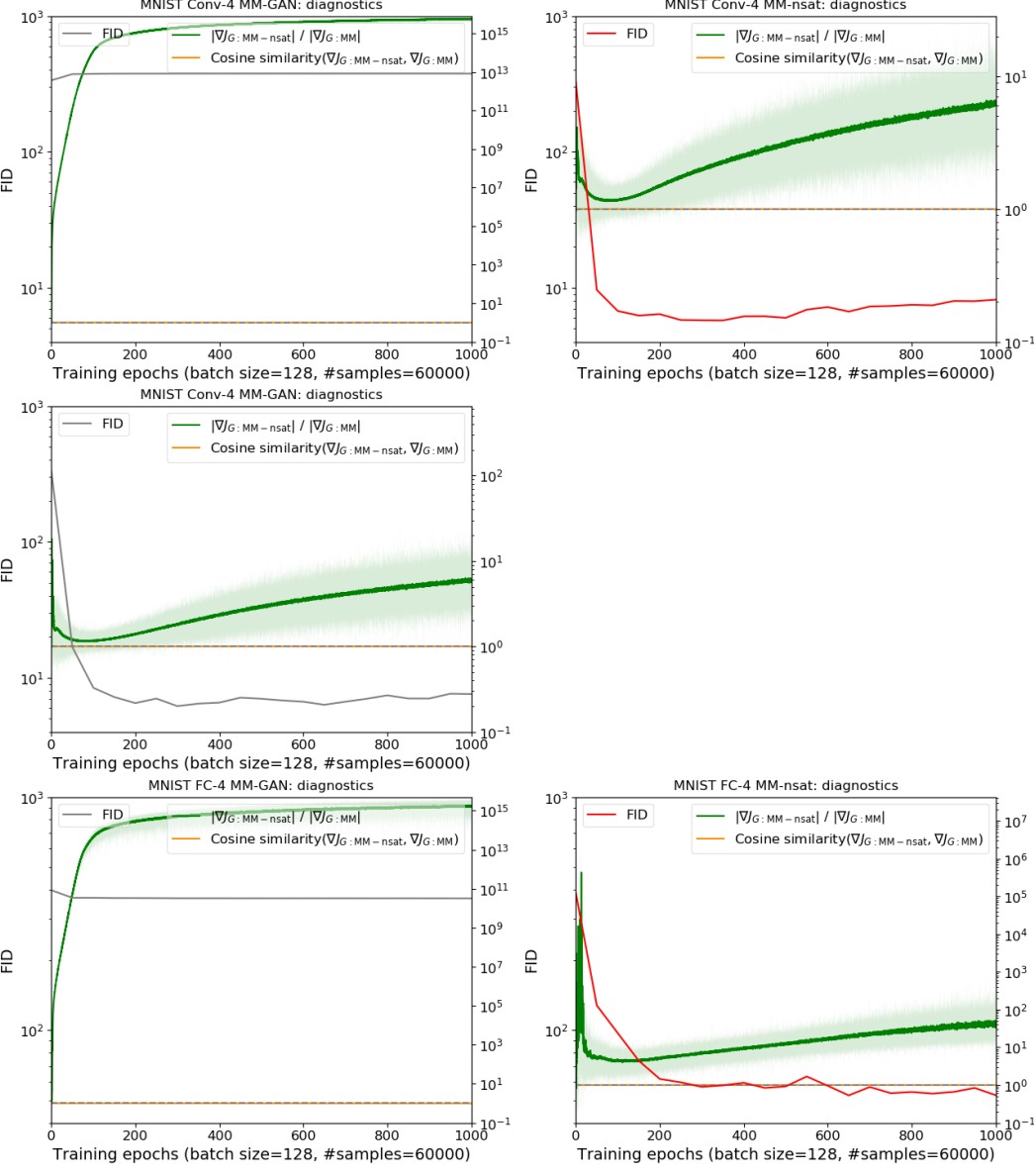

Figure 10: FID during training with additional diagnostics for 4-layer convolutional (top and middle) and fully-connected (bottom) networks on MNIST, using (left) MM-GAN and (right) MM-nsat for the parameter updates. For each run, we calculate both the MM-GAN and MM-nsat gradients at each iteration and plot their relative norms (green) and cosine similarity (orange). We plot the running means, with the actual, noisy values indicated by the shaded areas. Since both Conv-4 MM-GAN produces both good and bad results frequently, we show diagnostics for both cases.

## L    ADDITIONAL SATURATION RESULTS AND DISCUSSION

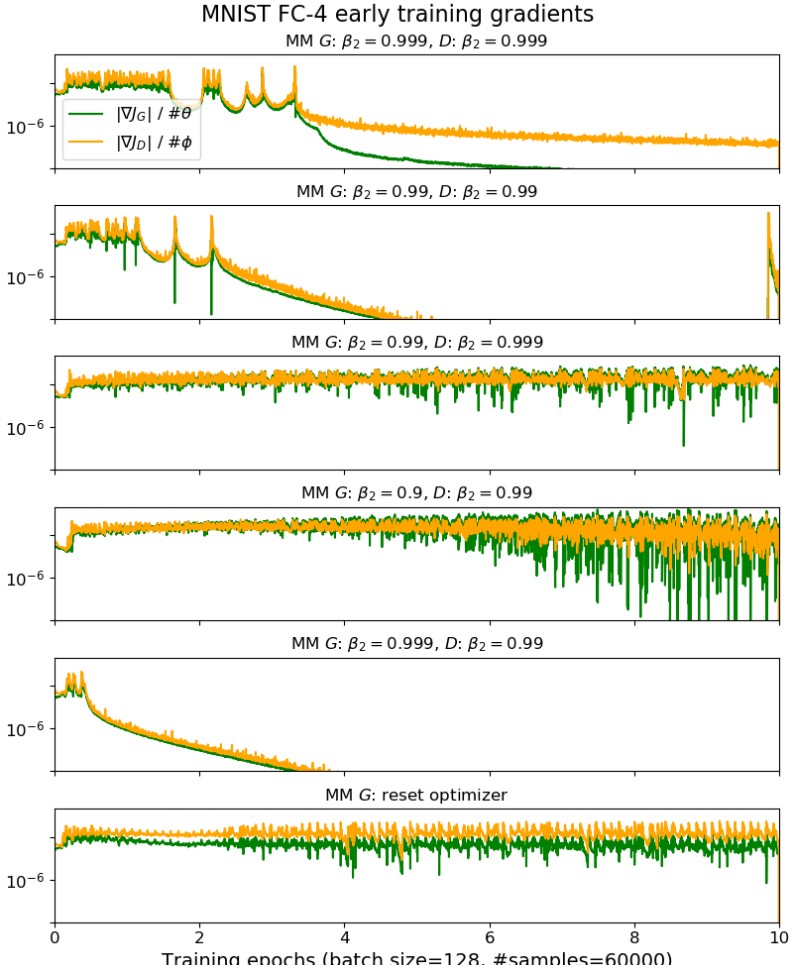

Figure 11: Gradient magnitudes in the early phase of training on MNIST with 4-layer fully connected networks: as in fig 5, but including additional tests for MM-GAN with modified $\beta_2$ hyperparameters for $D$'s and $G$'s optimizers. As suggested by our model in section 2.4, MM-GAN saturation when using Adam depends on $\beta_2$ values causing gradients to vanish. In particular, a lower value of $\beta_2$ for $G$ than $D$ disrupts the feedback mechanism that otherwise brings training to a halt.

When $D$ learns its task faster than $G$, $D_p(G(z))$ will begin to approach 0, such that both $D$'s and $G$'s gradients diminish. The effect on updates after Adam's normalization step depends on whether the first or second order momentum diminishes faster. The usual case is that $\beta_2 > \beta_1$, such that the second order momentum has a longer effective memory. This partially disables normalization of updates.

For $D$, this works as intended: as the cost approaches its minimum, the optimizer sees the gradients becoming smaller and reduces its step size, allowing proper convergence. The problem when optimizing MM-GAN with Adam is that this same behavior triggers for the ill-behaved $G$ cost function when it is failing at its task. While the update steps go in the right direction, the diminishing step size effectively causes parameters to get stuck in a poor configuration. Note that this problem does not really lie with Adam: it is the MM-GAN cost function that has a shortcoming that Adam cannot address since it only sees the decreasing gradients and has no way of knowing that this actually means that the MM-GAN cost is *increasing*. The role of the gradient rescaling in equation 6 is to make it so that $G$'s gradients and cost increase and decrease in parallel, such that the optimizer behaves properly.

### L.1 THE ADAM $\beta_2$ HYPERPARAMETER

Here, we provide additional empirical results (figure 11). In particular, we investigate the importance of how quickly $D$ and $G$'s updates vanish, relative to *each other*, by adjusting the $\beta_2$ parameters of $D$ and $G$ (our default is $\beta_2 = 0.999$). After all, the theory in 2.4 assumes a gradient that is simply exponential in time and does not model the interactions between $D$ and $G$ when the generated data gradients begin to vanish.

A fairly straightforward pattern emerges: for the standard case, where $D$ and $G$ have the same value of $\beta_2$, there are multiple $U$-shaped dips, where gradients alternate between diminishing and recovering, until the gradient finally vanishes for good. This still holds when when we reduce $\beta_2$ for both optimizers, effectively maintaining their balance. However, both setups where $\beta_2$ is lower for $G$ than for $D$ avoids vanishing updates entirely, whereas when we reduce the value of $\beta_2$ only for $D$, training breaks down earlier and more decisively. Finally, we show that we can avoid vanishing gradients by simply reinitializing $G$'s optimizer at regular intervals: this resets both the first and second order momenta. This stabilizes the training process, which is somewhat surprising seeing as we are effectively crippling the optimizer's memory. However, this effect is well explained by our hypothesis of large, lingering contributions to the second order momentum disabling normalization.

### L.2 LINEAR MODEL FOR $D_l(G(z))$

Figure 12 shows more detailed diagnostics for two runs where MM-GAN with standard Adam settings falls into its saturating failure mode. In addition to the gradients, we also plot the following: $\overline{D_l}$, the mean of the discriminator's logit outputs, both for real and generated data; and $|\Delta\theta|$ and $|\Delta\phi|$, the norm of the parameter update for $G$ and $D$ at each timestep.

In section 2.4, we assume that $D_l(G(z))$ behaves as a linear function of training steps. Note that this simple model is cannot fully capture the training dynamics for MM-GAN, but is meant to explain why Adam does not reliably address MM-GAN saturation despite its built-in gradient normalization.

The actual behavior of $D_l(G(z))$ in figure 12 is more complicated than we assume in our model. Particularly, $|D_l|$ grows much faster early on than it does towards the end, where it eventually tapers off. However, for $D_l(G(z)) = -20$, $D_p(G(z)) \approx 2 * 10^{-9}$, whereas $G$ only produces a total of $6 * 10^7$ samples during the 1000 epochs we use to train the networks. The behavior of $D_l$ past this point is of limited interest.

We find an early, *critical period* most relevant for understanding the interaction between Adam and the saturating MM-GAN cost, characterized by $D_l$ becoming monotonic for each of real and generated data. In figure 12, these critical periods start around 0.4 and 2.9 epochs for Conv-4 and FC-4 respectively, without any clearly defined endpoint. Within a single epoch, the step size for $G$'s updates drops by 3 to 4 orders of magnitude, and $G(z)$ ceases to change during training, as shown in figure 13.

The exception to the above is the sudden spike after 100 epochs for FC-4. Ideally, this spike would be the first step towards the training process recovering, as the size of $G$'s gradients and updates increases. However, while gradients are larger after this spike than before, the update step size actually plummets by an additional 4 orders of magnitude. The unusually large contribution to the second order momentum lingers much longer than the increase in the first order momentum, effectively disabling Adam normalization for a long while. This is a very illustrative example of the problematic interaction between the saturating MM-GAN cost and Adam.

### L.3 ADDITIONAL COMMENTS

Stabilizing MM-GAN by use of the $\beta_2$ parameter is not recommended: we intend these experiments to demonstrate the effects discussed in section 2.4. The setting where only $G$'s $\beta_2$ is reduced to 0.99 suffers the least from noisy updates and performs well in some settings, but all these variants are less stable than the more principled MM-nsat version. However, it should be noted that instead of changing the MM-GAN cost to work with Adam, as we do for MM-nsat, it is possible to modifiy the optimization procedure to account for MM-GAN's peculiarly shaped cost function.

NS-GAN owes some of its success to how $D$ and $G$ have different scaling factors. Only $D$'s gradients vanish as $D_p(G(z)) \to 0$, and only $G$'s gradients vanish as $D_p(G(z)) \to 1$. This gives the

training process a limited form of self-stabilization: update step sizes (roughly equivalent to learning rates) increase for whichever of the networks is doing worse. Since MM-nsat has the same gradient magnitudes as NS-GAN, this behavior carries over.

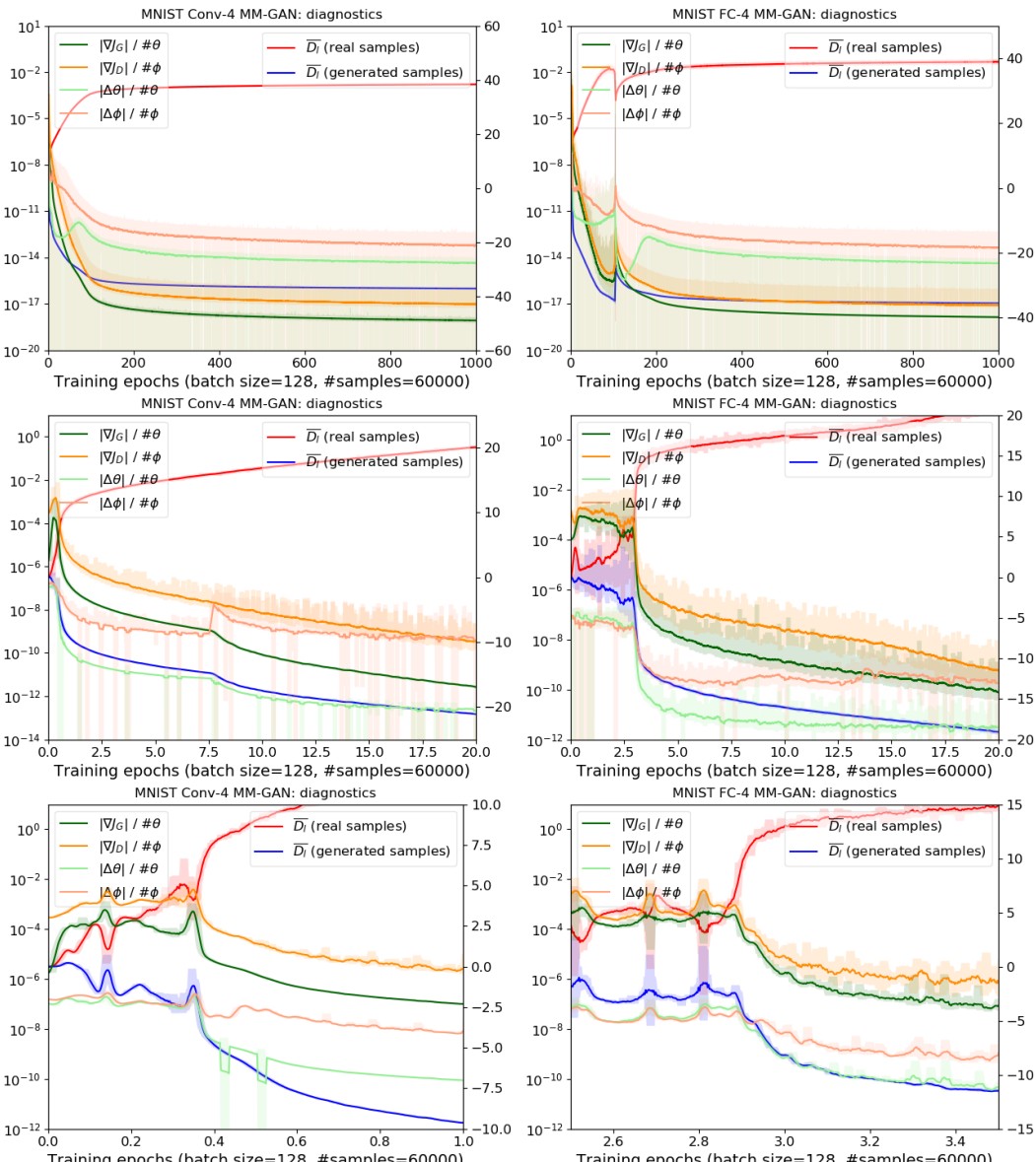

Figure 12: Gradient magnitudes (logarithmic) and mean logit outputs (linear) from $D$ during training for MNIST, using convolutional (left) and fully connected (right) networks. Top to bottom, plots show the same training run, increasingly closed up on the critical period where training breaks down due to gradients vanishing. We plot running means, with actual values indicated by shaded areas.

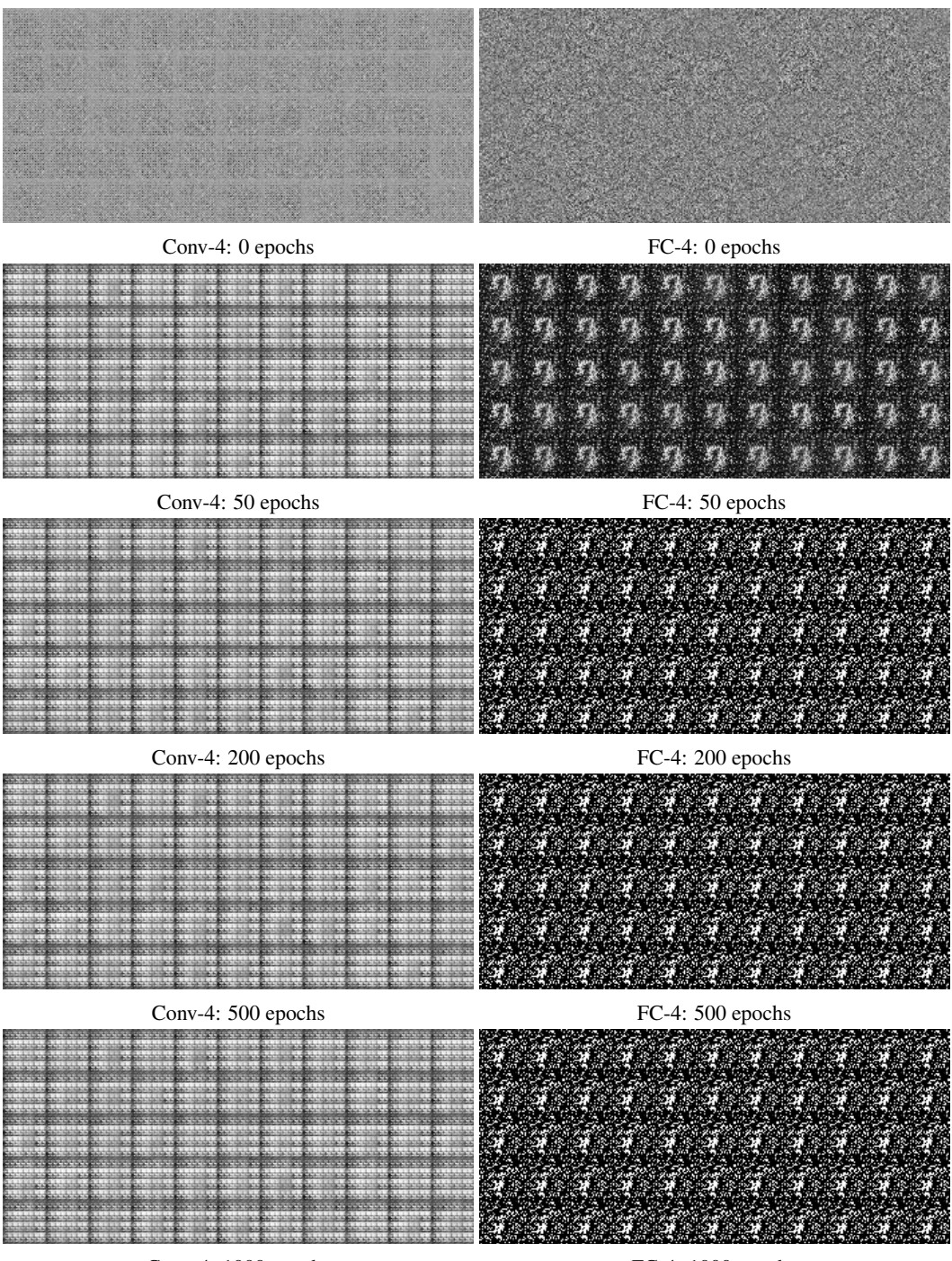

Figure 13: MNIST samples for 4-layer convolutional (left) and fully connected networks (right), using the saturating MM-GAN cost. Top to bottom show samples during training for a single run: we reuse the same set of latents $z$ to generate samples each time. Cross-reference with figure 12, showing diagnostics for the same runs. These samples show the behavior of the MM-GAN generator when it falls into its saturating failure mode: $G$'s outputs effectively cease to change due to negligible parameter updates.

# M    CIFAR-10 SPECTRAL NORMALIZATION BASELINE

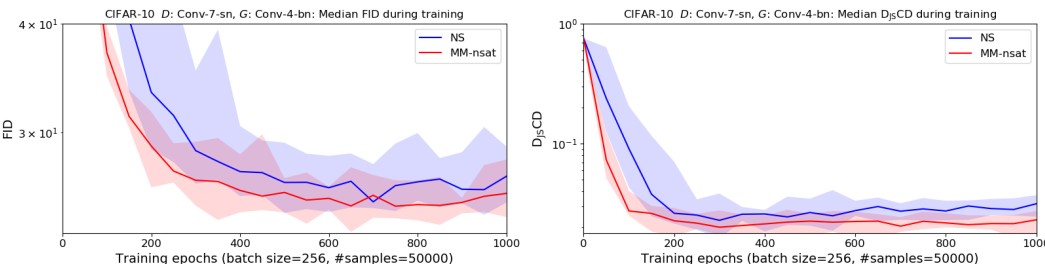

Figure 14: Left: FID, an estimated distance between generated and real data based on Inception activations. Right: $D_{JS}CD$, the Jensen-Shannon divergence between class distributions in generated and real data. Copying the CNN settings from Miyato et al. (2018) for the CIFAR-10 dataset, we plot the median values for ten runs during training. The shaded area indicates maximum and minimum values.

For the CIFAR-10 Conv-4-sn results at the bottom of figure 8, we use exceedingly simple networks as described in section 3.3. In order to compare MM-nsat against a benchmark from the literature, we the copy the settings from Miyato et al. (2018), i.e. the network architectures from **table 3** and the hyperparameter settings **C** from **table 1**.

The main differences between this setup and our Conv-4-sn settings are:

- Batch normalization in $G$
- 4x4-kernels (not 3x3) for 2-stride convolutions in $D$ and $G$
- 3 additional 3x3-kernel, 1-stride convolutions in $D$
- Spectral normalization also for the final layer of $D$
- Leaky ReLU in $D$
- Approximately 10 times as many network parameters
- Doubled learning rate

Additionally, for comparing FID directly against values tabulated in Miyato et al. (2018), note the following:

- Likely different batch size and training epochs
- We use all real and 50k generated images to estimate FID, instead of 10k and 5k

Across five runs, we obtain median FID values of 26.7 for NS-GAN and 25.5 for MM-nsat, using the same set of hyperparameters as Miyato et al. (2018) optimized for their NS-GAN results, reporting a value of 29.3. As in other experiments, we again see that $D_{JS}CD$ favors MM-nsat, even at the points during training where FID values are closely matched. Unlike for our networks, we see no clear tendency for the NS-GAN FID to increase towards the end of training, possibly due to higher quality of generated samples and spectral normalization also in the final layer of $D$ combining to pull $D_p$ values closer to $0.5$ throughout training. While there is significant overlap between the cost functions, MM-nsat shows slightly better performance overall and converges faster.

## N   Linear combinations of MM-GAN and NS-GAN

Motivated by theory connecting GAN divergences and mode dropping (section 2.3) and Huszár (2016) who suggests adding together the NS-GAN and MM-GAN cost functions in order to get rid of the subtracted Jensen-Shannon divergence for NS-GAN (eq 3), we run experiments using linear combinations of the cost function. Huszár's suggestion changes the scaling factor $(1 - D_p) \rightarrow 1$, which is an improvement in terms of the effect discussed in section 2.3, since it reduces the emphasis on overrepresented modes. However, if $D_p(G(z))$ is close to zero during training, as often happens with GANs since $D$ has an *easier* task than $G$, there will be little difference in practice.

We use the following linear interpolations of the MM-GAN and NS-GAN cost functions, parametrized by $a$ which determines the weight of each term:

$$
\begin{aligned}
\nabla_\theta J_{G_{\text{NS+MM}}}(a) &= (1 - a) \cdot \nabla_\theta J_{G_{\text{NS}}} + a \cdot \nabla_\theta J_{G_{\text{MM}}} \\
\nabla_\theta J_{G_{\text{NS+MM-nsat}}}(a) &= (1 - a) \cdot \nabla_\theta J_{G_{\text{NS}}} + a \cdot \nabla_\theta J_{G_{\text{MM-nsat}}}
\end{aligned}
\tag{24}
$$

Due to the MM-GAN and NS-GAN scaling factors (see fig 2), we expect linear combinations of MM-GAN and NS-GAN to be dominated by the non-saturating term, since $D_p(G(z))$ tends to be small when training GANs unless $D$ is strongly regularized. On the other hand, using our modified MM-nsat approximately balances the gradient norm for each term, such that the behavior of linear combinations should correspond to the weighting of the NS-GAN and MM-nsat terms.

Figure 15 shows results using linear combinations of NS-GAN, MM-GAN and MM-nsat for $G$'s cost. Results for NS-GAN and MM-nsat are as shown in the main text in figure 8. The behavior for NS-GAN and MM-nsat interpolations is most straightforward, with numerical results falling inbetween the pure NS-GAN and MM-nsat variants with roughly the degree of separation suggested by their weighting constants. Since NS-GAN and MM-nsat have approximately equal gradient magnitudes for *all* values of $D_p(G(z))$, the contributions from each term are predictable.

For NS-GAN and MM-GAN interpolations, behavior is less consistent and more similar to NS-GAN. This is reasonable, seeing as NS-GAN and MM-GAN have scaling factors $1 - D_p$ and $D_p$ respectively, such that the two terms are effectively weighted both explicitly by the weighting constant and implicitly by whatever the values of $D_p(G(z))$ happen to be during training. Note that if $D$ completely fails to discriminate between real and generated samples, it can minimize its cost by making $D_p(x) = D_p(G(z)) = 0.5$, such that NS-GAN's scaling factor will almost always be larger than that of MM-GAN.

For the FC-4 results, note that even a $\frac{9}{10}$ weighting for MM-GAN still gives behavior qualitatively similar to NS-GAN: the early stopping metrics are much better than for pure NS-GAN and the mode collapsing behavior is delayed, but the final metrics are worse both in terms of FID and $D_{\text{JS}}$CD even than for the $\frac{1}{3}$-weighted MM-nsat. Generally, Huszár's suggestion of adding together the NS-GAN and MM-GAN leads to somewhat improved results, but does not come close to stability and mode coverage achieved by MM-nsat. Compare also to the divergences shown in eq 3.

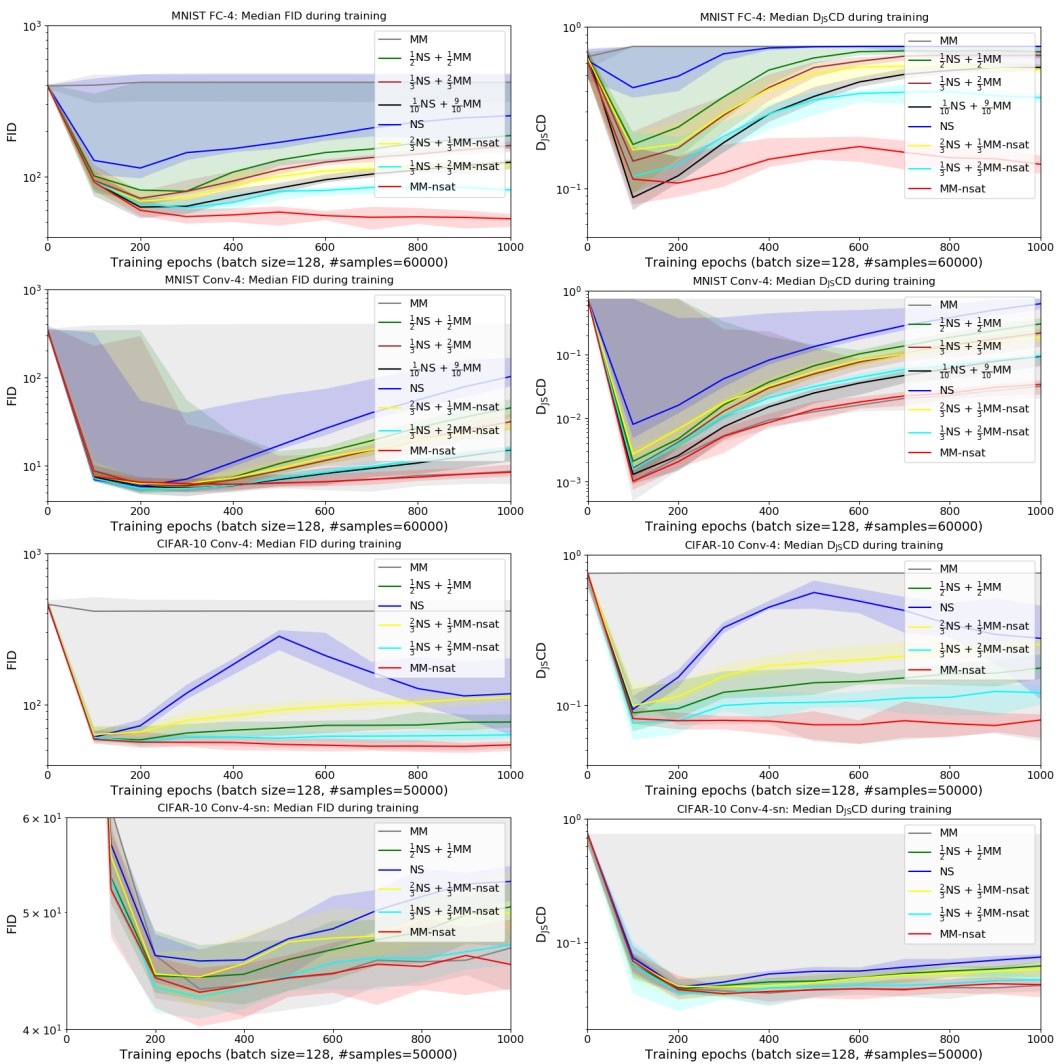

Figure 15: Left: FID, an estimated distance between generated and real data based on Inception activations. Right: D$_{JS}$CD, the Jensen-Shannon divergence between class distributions in generated and real data. For various networks using the MNIST and CIFAR-10 datasets, we plot the median values for ten runs during training. The shaded area indicates maximum and minimum values. Costs are the traditional MM-GAN and NS-GAN (Goodfellow et al., 2014), as well as our MM-nsat from eq 6 and linear combinations as given in eq 24.

## O   SCALING FACTOR EXPERIMENTS

In section 2.1, we show that the gradients for the MM-GAN and NS-GAN generators have different scaling factors:

$$\nabla_\theta J_{G_{\text{MM}}} = -\frac{\partial D_l(G(z,\theta))}{\partial \theta} \cdot D_p(G(z,\theta))$$
$$\nabla_\theta J_{G_{\text{NS}}} = -\frac{\partial D_l(G(z,\theta))}{\partial \theta} \cdot (1 - D_p(G(z,\theta))) \tag{25}$$

In section 2.3, we suggest that the different emphasis this places on under- and oversampled modes is the reason for the more zero-avoiding and mode-covering behaviors of NS-GAN and MM-GAN. To test this hypothesis, we can modify the scaling factors and see whether we get the expected change in behavior. The linear interpolations used in section N probe the same effect in a less direct manner.

While we expect the importance of the scaling factors to depend on the relative weights they assign to different scoring samples, it is not clear which relationships are most crucial. The difference between MM-GAN and NS-GAN is most pronounced for low- and high-scoring samples, but the important difference might very well be the relative weights of two samples with different, low scores.

A simple way to modify the NS-GAN gradient is to add a constant $a$ to the scaling factor:

$$\nabla_\theta J_{G_{\text{NS-add-a}}} = -\frac{\partial D_l(G(z,\theta))}{\partial \theta} \cdot (1 - D_p(G(z,\theta)) + a) \tag{26}$$

Which is obtained by the following single-sample cost function:

$$J_{G_{\text{NS-add-a}}} = J_{G_{\text{NS}}} - a * D_l \tag{27}$$

Renormalizing the total gradient magnitude in the same way as for MM-nsat, we get the following cost function:

$$\nabla_\theta J_{G_{\text{NS-add-a}}}^{\text{batch}} = \frac{1 - \overline{D_p}}{a + 1 - \overline{D_p}} \sum_{i=0}^{N-1} \nabla_\theta (J_{G_{\text{NS}}}(z_i) - a * D_l(z_i)) \tag{28}$$

And similarly for MM-GAN:

$$\nabla_\theta J_{G_{\text{MM-nsat-add-a}}}^{\text{batch}} = \frac{1 - \overline{D_p}}{a + \overline{D_p}} \sum_{i=0}^{N-1} \nabla_\theta (J_{G_{\text{MM}}}(z_i) - a * D_l(z_i)) \tag{29}$$

These cost functions are only reasonable for $a \geq 0$, such that we avoid negative scaling factors for samples. For increasingly large values of $a$, sample weights effectively become uniform across the whole range of possible scores.

Since these modifications only allow us to make MM and NS less extreme and more similar to each other, we also try a different approach, introducing a exponentiation parameter $c$ for the scaling factor:

$$\nabla_\theta J_{G_{\text{NS-exp-c}}} = -\frac{\partial D_l(G(z,\theta))}{\partial \theta} \cdot (1 - D_p(G(z,\theta)))^c \tag{30}$$

The general solution of this differential equation can be expressed in terms of the hypergeometric function, which does not lend itself to efficient computation. However, specific values of $c$ give rise to useful cost functions. As before, we consistently renormalize the gradient magnitude to that of NS-GAN:

$$J_{G_{\text{NS-exp-2}}}^{\text{batch}} = \frac{1}{1 - \overline{D_p}} \sum_{i=0}^{N-1} \nabla_\theta (J_{G_{\text{NS}}}(z_i) - D_p(z_i)) \tag{31}$$

$$J_{G_{\text{MM-nsat-exp-2}}}^{\text{batch}} = \frac{1 - \overline{D_p}}{\overline{D_p}^2} \sum_{i=0}^{N-1} \nabla_\theta (J_{G_{\text{MM}}}(z_i) + D_p(z_i)) \tag{32}$$

$$J_{G_{\text{NS-exp-0.5}}}^{\text{batch}} = (1 - \overline{D_p})^{\frac{1}{2}} \sum_{i=0}^{N-1} \nabla_\theta (2 \log(\sqrt{e^{-D_l}} + \sqrt{e^{-D_l} + 1})) \tag{33}$$

$$J_{G_{\text{MM-nsat-exp-0.5}}}^{\text{batch}} = \frac{1 - \overline{D_p}}{\overline{D_p}^{\frac{1}{2}}} \sum_{i=0}^{N-1} \nabla_\theta (-2 \log(\sqrt{e^{D_l}} + \sqrt{e^{D_l} + 1})) \tag{34}$$

Renormalizing the overall gradient magnitude goes a long way towards stabilizing the adversarial training dynamics for these cost functions. However, overemphasis on oversampled modes tends to accelerate catastrophical mode collapse. Furthermode, expressions such as $\log(\sqrt{e^{D_l}}))$ are numerically unstable for large values of $|D_l|$.

Results using these variant cost functions are shown in figure 16. The *add*-variants have very clean behavior, all falling in between NS and MM-nsat in terms of FID and $D_{\text{JS}}$CD, in the same order as suggested by their more uniform scaling factors. The *exp*-variants are more erratic, with NS-exp-2 having major stability issues and MM-nsat-exp-2 falling off later in training for CIFAR-10 Conv-4. Aside from these points, results correspond with our theoretical expectations.

Perhaps the most striking result is the strong performance of the MM-nsat-exp-2 cost function. This strange variant is designed simply to have a more extreme version of the minimax scaling factor, which we expect to further temper the the mode-dropping mechanism described in section 2.3. Indeed, it generally improves performance relative MM-nsat.

Finally, we note that all the MM-nsat-add variants tend towards stronger mode collapse than the MM-nsat-exp variants, regardless of the choice of parameter. We suggest the following explanation for this behavior: Consider two generated samples, both such that $D_p(G(z))$ is close to 0. With NS sample weighting or with MM-nsat-add variants, the relative weights of these samples will be close to 1, simply because a small $D_p$ is negligible relative to the additive constants. With MM sample weighting and MM-nsat-exp variants, on the other hand, relative weights have a strong dependence the exact values of $D_p$ for each sample and may be orders of magnitude apart.

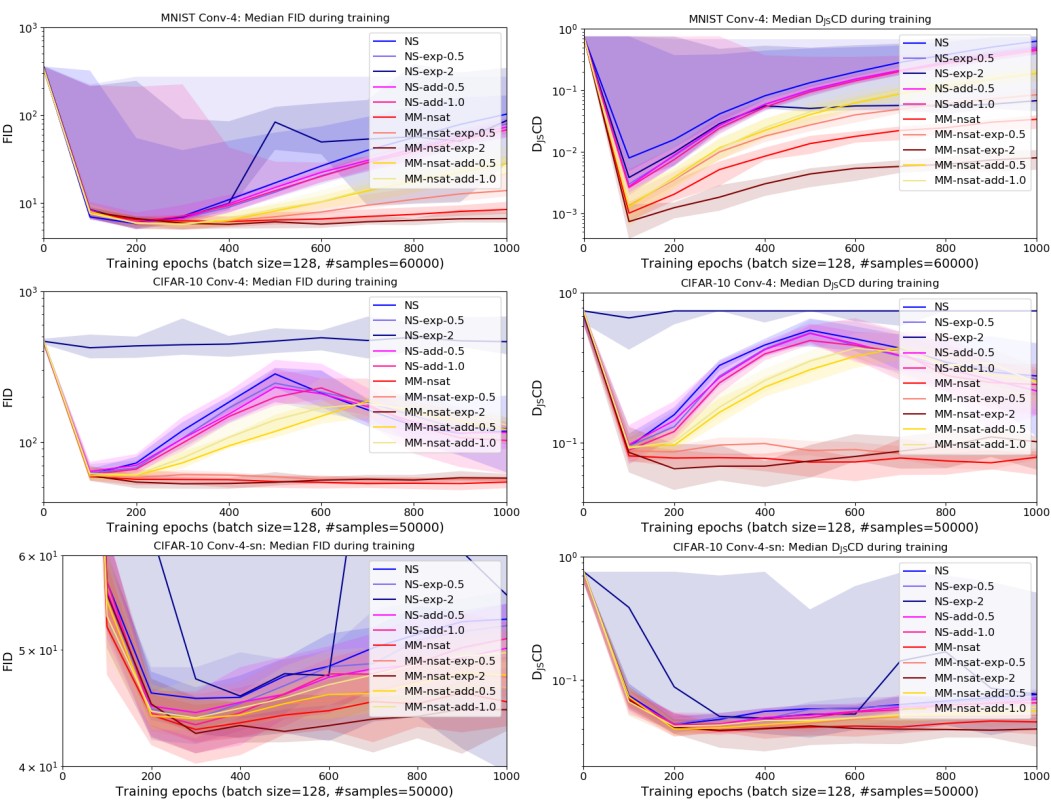

Figure 16: Left: FID, an estimated distance between generated and real data based on Inception activations. Right: $D_{JS}CD$, the Jensen-Shannon divergence between class distributions in generated and real data. For various networks using the MNIST and CIFAR-10 datasets, we plot the median values for ten runs during training. The shaded area indicates maximum and minimum values. Modified cost functions behave as predicted by their scaling factors according to theory in section 2.3. In particular, MM-nsat-exp-2, using a scaling factor with even more emphasis on underrepresented modes, achieves better values of FID and $D_{JS}CD$ than the more standard cost functions.

## P   HINGE-GAN AND LS-GAN

To supplement our results for NS-GAN and MM-GAN variants in section 4.2, we choose to run experiments with the LS-GAN (Mao et al., 2016) and Hinge-GAN (Miyato et al., 2018) formulations. As these only require the cost functions themselves to be modified, they can be compared directly to NS-GAN and MM-nsat. Note that we do not adjust network architectures or hyperparameters from the default settings we use across all experiments, such that results are more indicative of general stability than best-case performance.

The widely used WGAN (Arjovsky et al., 2017) formulation requires additional regularization to prevent $D$'s outputs from diverging, most commonly a 1-centred gradient penalty on interpolations between real and generated data (Gulrajani et al., 2017). This makes it difficult to draw fair comparisons. Hinge-GAN is very similar to WGAN, but simply clips $D$'s cost outside of the $(0, 1)$ interval and is known to produce strong results in for instance BigGAN (Brock et al., 2018).

### P.1   MNIST AND CIFAR-10 RESULTS

Figure 17 shows results, as in figure 8 in the main text, but with a different set of cost functions. Figure 1 includes a subset of these plots. For the bottom figure, note the use of spectral normalization and a much smaller interval of FID values than the other plots. As for NS-GAN and MM-nsat, we have not adjusted networks

Generally speaking, Hinge-GAN performs better than NS-GAN but worse than MM-nsat and suffers from some of the same gradual mode collapse issues as NS-GAN. Where Hinge-GAN and MM-nsat are most similar in terms of FID, MM-nsat tends towards better class balance. LS-GAN performs remarkably well for MNIST Conv-4, but is otherwise unimpressive and suffers the most from stability issues.

### P.2   GRADUAL MODE COLLAPSE FOR HINGE-GAN AND LS-GAN

Interestingly, the results in figure 17 suggest that the behavior of gradual mode collapse seen for NS-GAN and explained in section 2.3 is not specific to NS-GAN. This is most clear for Hinge-GAN, which in a qualitative sense follows the same trajectory as NS-GAN in terms of both FID and $D_{JS}CD$, but with better numerical results overall. For LS-GAN, we see similar tendencies, but its behavior is much more erratic overall, including very poor early stopping results for MNIST FC-4 and very strong results for CIFAR-10 Conv-4, even beating MM-nsat.

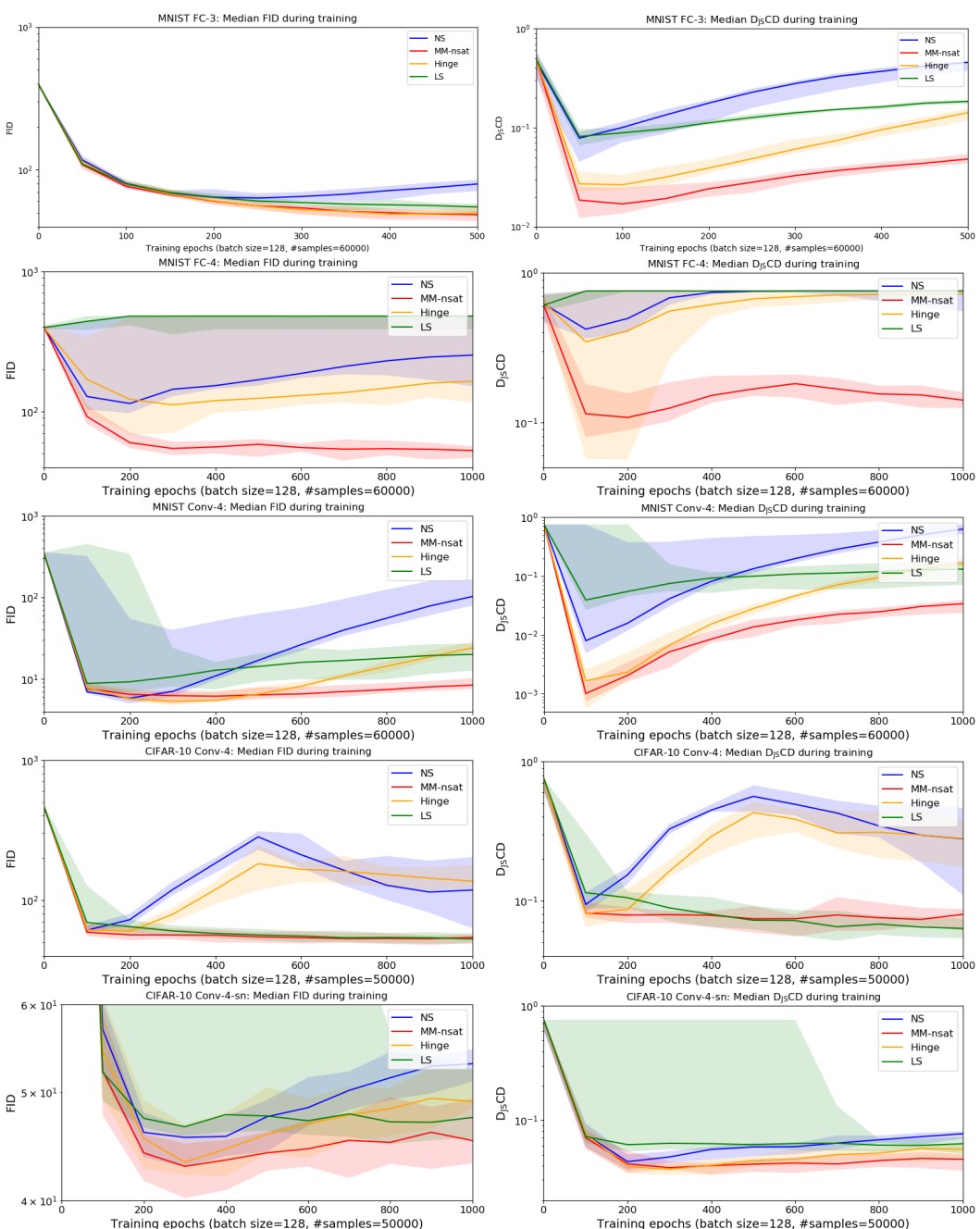

Figure 17: Left: FID, an estimated distance between generated and real data based on Inception activations. Right: D$_{JS}$CD, the Jensen-Shannon divergence between class distributions in generated and real data. For various networks using the MNIST and CIFAR-10 datasets, we plot the median values for ten runs during training. The shaded area indicates maximum and minimum values. NS is Goodfellow's non-saturating cost; MM-nsat is from eq 6; Hinge and LS are alternative formulations for comparison, which use different costs for both $D$ and $G$.

## Q    SAMPLE WEIGHTING AND MODE DROPPING FOR OTHER GAN FORMULATIONS

In section 2.3, we suggest that the mode dropping tendencies of NS-GAN can be understood as a consequence of how overrepresentation and upweighting interact, allowing generated samples from a small region of the data space to dominate $G$'s parameter updates. In this section, we discuss the wider implications of this effect.

Note in particular the following figures showing relevant empirical results, with additional discussion in the subsections:

- MM-GAN and NS-GAN variants: fig 8
- MM-GAN and NS-GAN linear combinations: fig 15
- MM-GAN and NS-GAN scaling factor modifications: fig 16
- Hinge-GAN and LS-GAN comparisons: fig 17 and fig 21

The key result from these experiments is that the relationship between upweighting of overrepresented samples and gradual mode dropping is robust across a variety of changes in sample weighting, both implicitly (linear combinations) and explicitly (scaling factor modifications). Increasingly upweighting scaling factors give increasingly mode dropping behavior. Furthermore, Hinge-GAN and LS-GAN suffer from mode dropping behavior which is similar to that of NS-GAN in a qualitative sense: this is consistent for Hinge-GAN, while LS-GAN is more erratic and suffers from stability issues.

In section 2.3 we only discuss the scaling factors of NS-GAN and MM-GAN, which respectively upweight and downweight overrepresented samples. To understand the effect of scaling factors in general, it is instructive to consider the linear combination $\frac{1}{2}$NS $+ \frac{1}{2}$MM in figure 15. Its scaling factor is simply $\frac{1}{2}(1 - D_p(G(z))) + \frac{1}{2}D_p(G(z)) = \frac{1}{2}$: in other words, it neither upweights or downweights samples.

Despite its uniform sample weighting, the linear combination $\frac{1}{2}$NS $+ \frac{1}{2}$MM still has mode dropping tendencies, though it is less blatant than for NS-GAN. This is expected from our theory in section 2.3: MM-GAN downweights overrepresented samples to compensate for their overrepresentation. Uniform weighting, while better than NS-GAN upweighting, does not avoid the mode dropping effect, because overrepresentation on its own is enough to cause gradual mode dropping.

While it is difficult to make general claims about how sample weighting relates to mode dropping for other GAN formulations in general, we can consider Hinge-GAN which has simple cost functions: its generator has the same uniform sample weighting as the linear combination $\frac{1}{2}$NS $+ \frac{1}{2}$MM discussed above, and its discriminator *also* has clipped, uniform sample weighting, unlike the error-emphasizing cross entropy loss for MM-GAN and NS-GAN. While the Hinge-GAN discriminator will behave differently during training, it does not give Hinge-GAN any mechanism for downweighting overrepresented samples, explaining the mode dropping behavior seen in our results.

MM-GAN's form of downweighting is highly unusual. Note that we mean overrepresentation of generated samples *relative to real samples*: MM-GAN downweights realistic samples that are generated too often. However, it also downweights highly unrealistic samples, since these will be overrepresented by virtue of falling in regions where the density of real samples is negligible. This downweighting is a liability early in training, because it tends to cause saturation problems when $D$ learns to assign very low values of $D_p$ to $G$'s unrealistic samples. Note results in section O, which indicate that strong downweighting is necessary to stabilize the overrepresentation effect.

This means that cost functions which properly downweight overrepresented samples will generally fail to train unless care is taken to decouple overall gradient magnitude (determining saturation) from sample weighting (determining mode dropping).

## R   Discriminator regularization

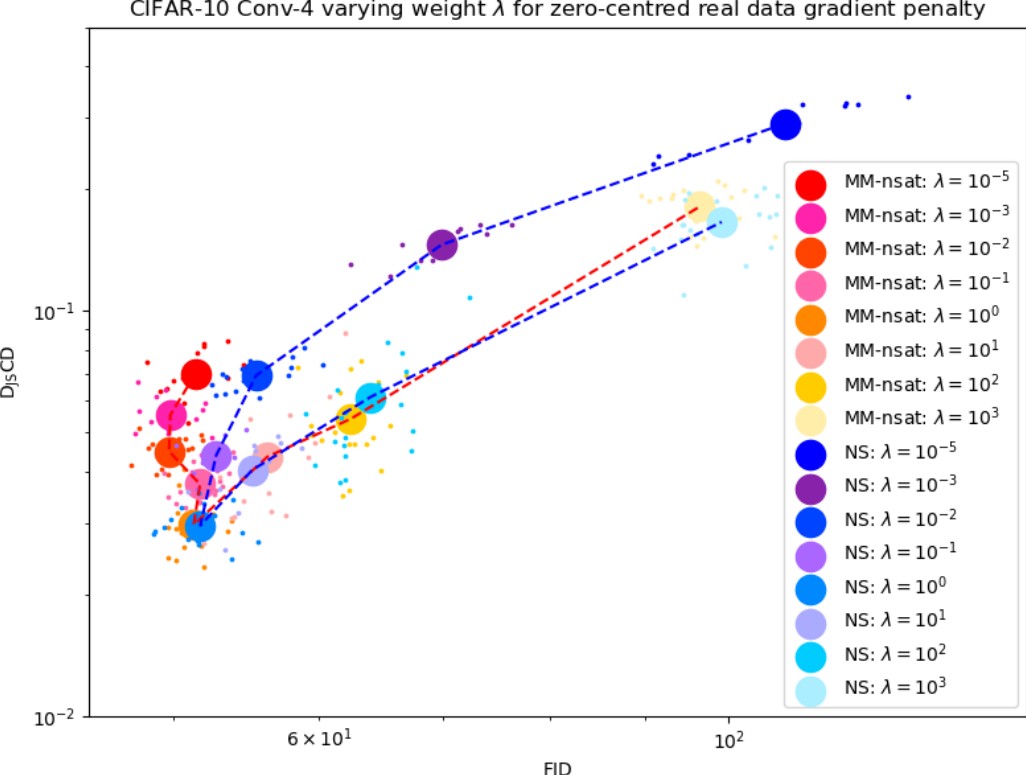

Figure 18: Scatter plot $D_{JS}CD$ vs FID for 4-layer convolutional networks on CIFAR-10, using MM-nsat and NS cost functions. We regularize $D$ with zero-centred real data gradient penalty, sweeping across different values of the weighting constant $\lambda$ used in $D$'s cost function, and train for 1000 epochs. Each dot represents a single run, while the larger circles show the mean value for the twenty different runs for each value of $\lambda$. The dotted lines indicate each cost function's *trajectory* for increasing values of $\lambda$. With very weak regularization, MM-nsat is far stronger than NS, but for increasingly strong regularizations, the two costs converge in terms of performance. Their best results ($\lambda = 10^0$) are similar and match weakly regularized MM-nsat in terms of FID, but with improved class balance. Performance progressively degrades with stronger regularization.

Figure 18 shows results for MM-nsat and NS cost functions when combined with zero-centred real data gradient penalty (Mescheder, 2018). See also results for spectral normalized networks at the bottom in figures 8 and 17, where results are much more similar than in the unregularized case: this still holds true with different cost functions and a different form of regularization. This is essentially a special case of an effect shown in Qin et al. (2018), that all strongly regularized cost functions degenerate into the same behavior.

The key result is how strong regularization makes the performance of MM-nsat and NS (and other cost functions) much more similar. Some form of regularization is often used to obtain strong results, since the introduction of various forms of gradient penalties such as the one used in WGAN-GP (Gulrajani et al., 2017) and spectral normalization (Miyato et al., 2018).

Discriminator regularization has many effects, such as reducing the ability of $D$ to learn quickly and smoothing the landscape of $D$'s outputs. It is difficult to fully understand how it affects the training process for GANs. Generally speaking, limiting the Lipschitz value of $D$ (as is done explicitly by spectral normalization and partially by zero-centred real data gradient penalty) flattens the shape of $D$ and ultimately limits the difference between the outputs for real and generated data. It follows that assuming that $D \approx D^{\text{opt}}$ as given by eq 7 is less reasonable for a regularized discriminator, such that the effect discussed in section 2.3 will be less pronounced.

As discussed in section 2.1, the difference between MM-GAN and NS-GAN is only in the scaling factors, $D_p$ and $1 - D_p$ respectively. Strong regularization effectively squeezes $D_p$ towards 0.5: this makes $D_p$ and $1 - D_p$ more similar, and thus also MM-nsat and NS, as seen in fig 18. For the runs we have plotted, regularization vastly improves the results for NS, but the benefits for MM-nsat are much more limited. The common default value of the weighting for the gradient penalty, $\lambda = 10^1$, is actually somewhat harmful for MM-nsat compared to the unregularized case, and looking only at values for this regularization gives the misleading impression that NS is stronger than MM-nsat.

We consider a more thorough discussion of this problem out of scope for this work, but suggest that it might explain the fairly similar results for MM-nsat and NS when applied to the StyleGAN model (see W). Absent a better understanding of the interactions between discriminator regularization and cost functions, hyperparameters for regularization terms should be tested extensively when applying MM-nsat.

## S  RING OF GAUSSIANS

The *ring of Gaussians* toy problem has a number of degrees of freedom, both for the dataset (number of modes, number of standard deviations of separation) and the model (architecture, batch size, training iterations). There are settings where both MM-GAN and NS-GAN cover or drop modes and significant variation between individual runs. We do not present a thorough analysis of this problem: in the main text, we use settings chosen to obtain the same qualitative NS-GAN behavior as shown in Metz et al. (2016); Srivastava et al. (2017).

To give some impression of the diversity of possible behaviors and the usual differences between NS-GAN and MM-GAN, we include results for a more challenging toy problem in figure 19 and table 3. In this case, where NS-GAN drops only some modes, we again find that MM-GAN has qualitatively better mode coverage. Furthermore, frequencies of samples from each mode is much better aligned with real data for MM-GAN than for NS-GAN. Interestingly, while NS-GAN places particular emphasis on avoiding generated samples outside of the real data manifold, *more* samples fall outside of the real data modes for NS-GAN than for MM-GAN.

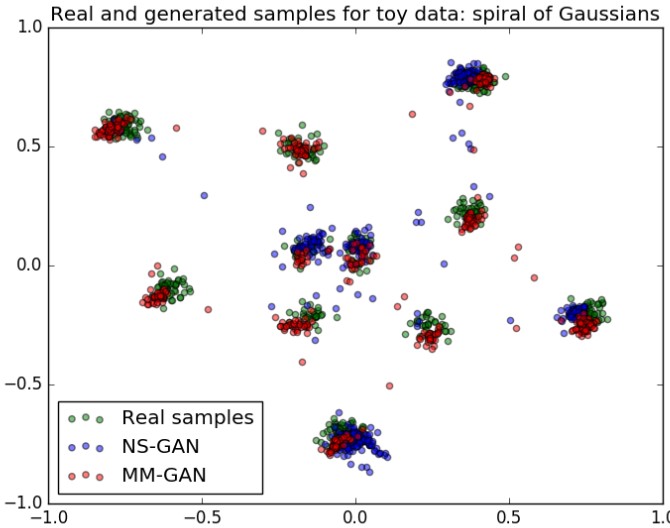

Figure 19: Real and generated samples on modified 2D mixture of Gaussians: modes ordered along an expanding spiral from the origin, making in excess of two revolutions. Density of modes increases along this spiral. In blue, NS-GAN. In red, samples using the unmodified MM-GAN cost function. MM-GAN trains well on this toy problem despite its saturating cost function and shows decent coverage of all modes. Cross-reference with table 3.

Table 3: Densities for each mode for generator with 3-layer fully connected networks. Samples are classified as belonging to the closest real data mode if it is within 3 standard deviations of any mode, otherwise as belonging to no mode.

| | **Mode frequency in %** | | | | | | | | | | | |
|---|---|---|---|---|---|---|---|---|---|---|---|---|
| **Mode** | *None* | 1 | 2 | 3 | 4 | 5 | 6 | 7 | 8 | 9 | 10 | 11 | 12 |
| **Cost** | | | | | | | | | | | | |
| NS | 4.3 | 1.4 | 8.6 | 12 | 1.0 | 0.6 | 0.6 | 0.0 | 0.0 | 25 | 19 | 27 | 1.2 |
| MM | 2.5 | 2.7 | 2.5 | 3.3 | 6.8 | 5.5 | 8.0 | 7.8 | 8.6 | 11 | 13 | 14 | 15 |
| *Data* | - | 2.0 | 3.1 | 4.3 | 5.4 | 6.6 | 7.7 | 8.9 | 10 | 11 | 12 | 13 | 15 |

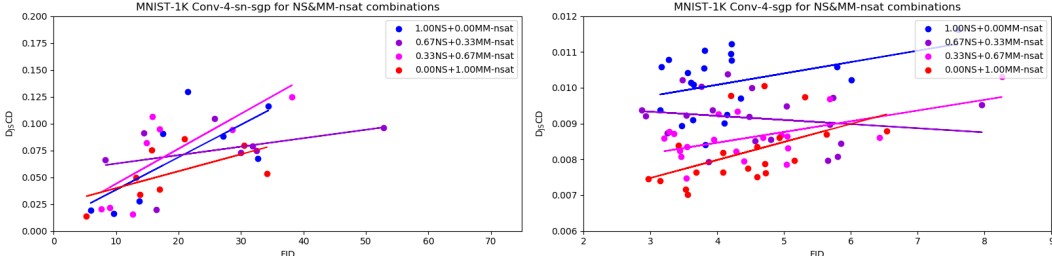

Figure 20: Scatterplots D$_{JS}$ vs FID with trendlines for 20 training runs, using linear combinations of the NS and MM-nsat costs (eq 24), on the MNIST-1K dataset created by combining triplets of samples from MNIST into 3-channel RGB images. Left: zero centred gradient penalty and spectral normalization. Right: no spectral normalization and relaxed zero centred gradient penalty.

## T    MNIST-1K

We run tests on the Stacked MNIST dataset (Metz et al., 2016) (also known as MNIST-1K), in order to get a more demanding and multi-modal training task where we can still classify generated samples. The samples in this dataset are obtained by combining three samples from MNIST to represent each of the three color channels for an RGB image. We find a variety of network architectures difficult to train for any of our cost functions and resort to regularizing $D$. We show results in figure 20: in both cases, differences between linear combinations are minor. For the most heavily regularized model, there is no clear trend. Removing spectral normalization and relaxing the zero centred gradient penalty improves the models and makes them slightly more different, recovering the usual ordering in terms of D$_{JS}$.

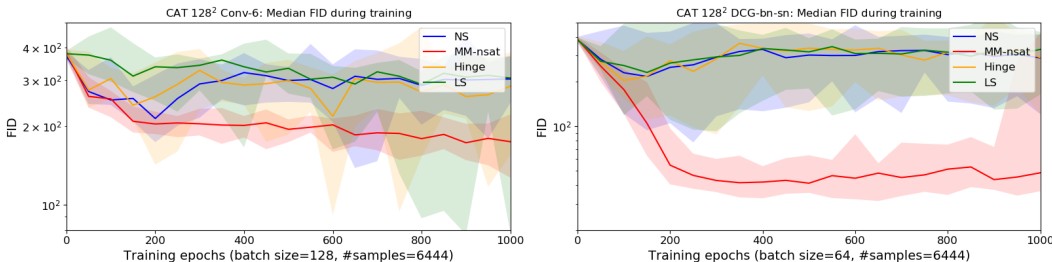

Figure 21: FID during training for ten runs on CAT $128^2$ with two different network architectures. Left: 6-layer convolutional networks. Right: DCGAN with batch normalization as well as spectral normalization for $D$. For both architectures, only MM-nsat trains reasonably well, while NS-GAN, Hinge-GAN and LS-GAN all have major stability issues due to early, catastrophical mode collapse.

## U   CAT $128^2$

We run tests with CAT $128^2$ (Zhang et al., 2008), chosen as a reasonably difficult dataset which is still amenable to simple network architectures. FID during training for two architectures is shown in figure 21, emphasizing training stability. Figures 22a and 22b show samples for DCGAN with spectral normalization for $D$ and self attention: these are shown cropped in the main text in figure 7. Figures 22c, 22d and 22e show samples for other network architectures. Note that samples are JPEG compressed.

We find that MM-unit and MM-nsat perform much better than NS in this setting. In particular, we find that the failure mode of NS cannot be addressed by early stopping in these experiments, unlike for MNIST, where NS tends to produce better results early on and mode dropping is mostly due to extended training (see figure 8). For CAT $128^2$, catastrophical mode collapse tends to happen before $G$ has learned to produce reasonable samples. MM-nsat is much less susceptible to this failure mode than NS. As usual, results are most similar with regularized discriminator.

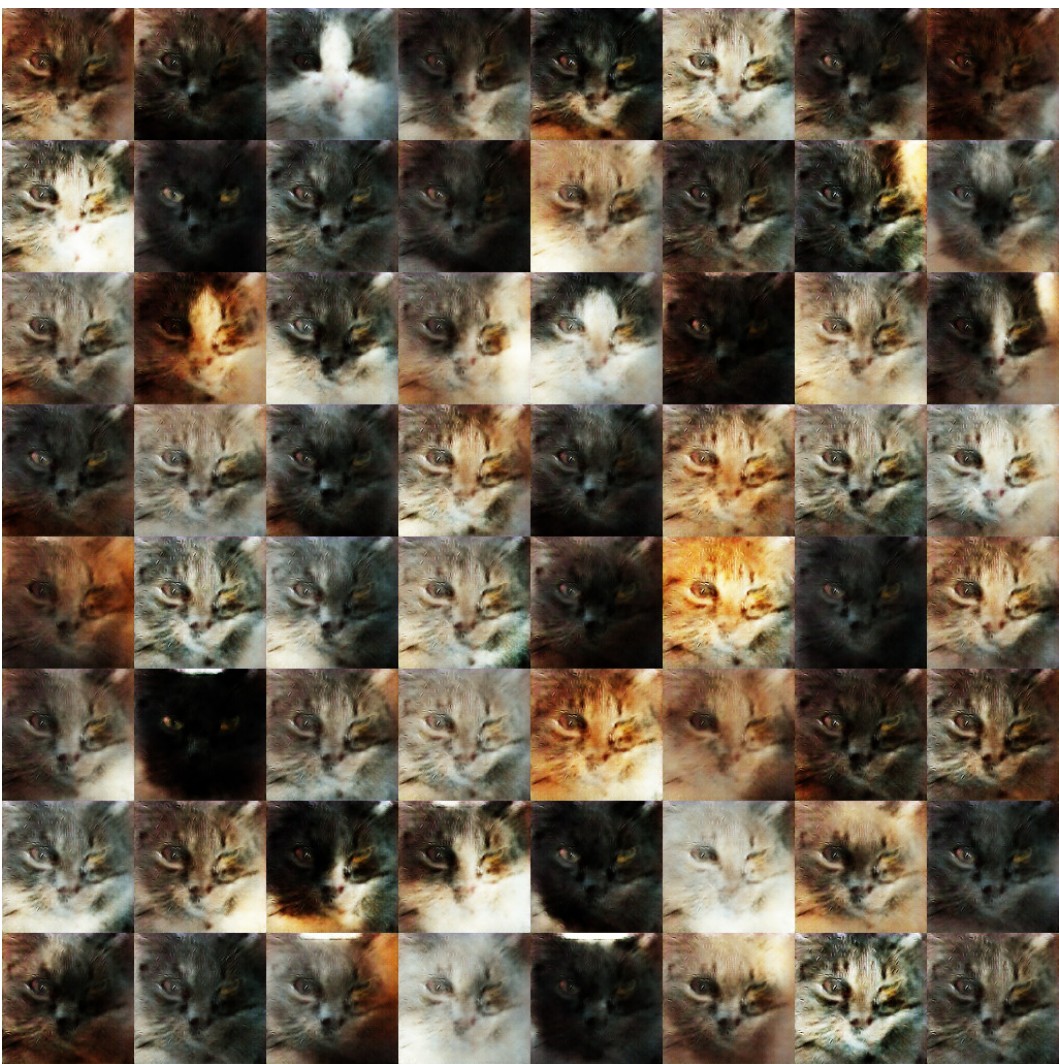

Figure 22a: Early stopping samples from best NS run for CAT $128^2$ using DCGAN with spectral normalization in the discriminator and self-attention (as in main text).

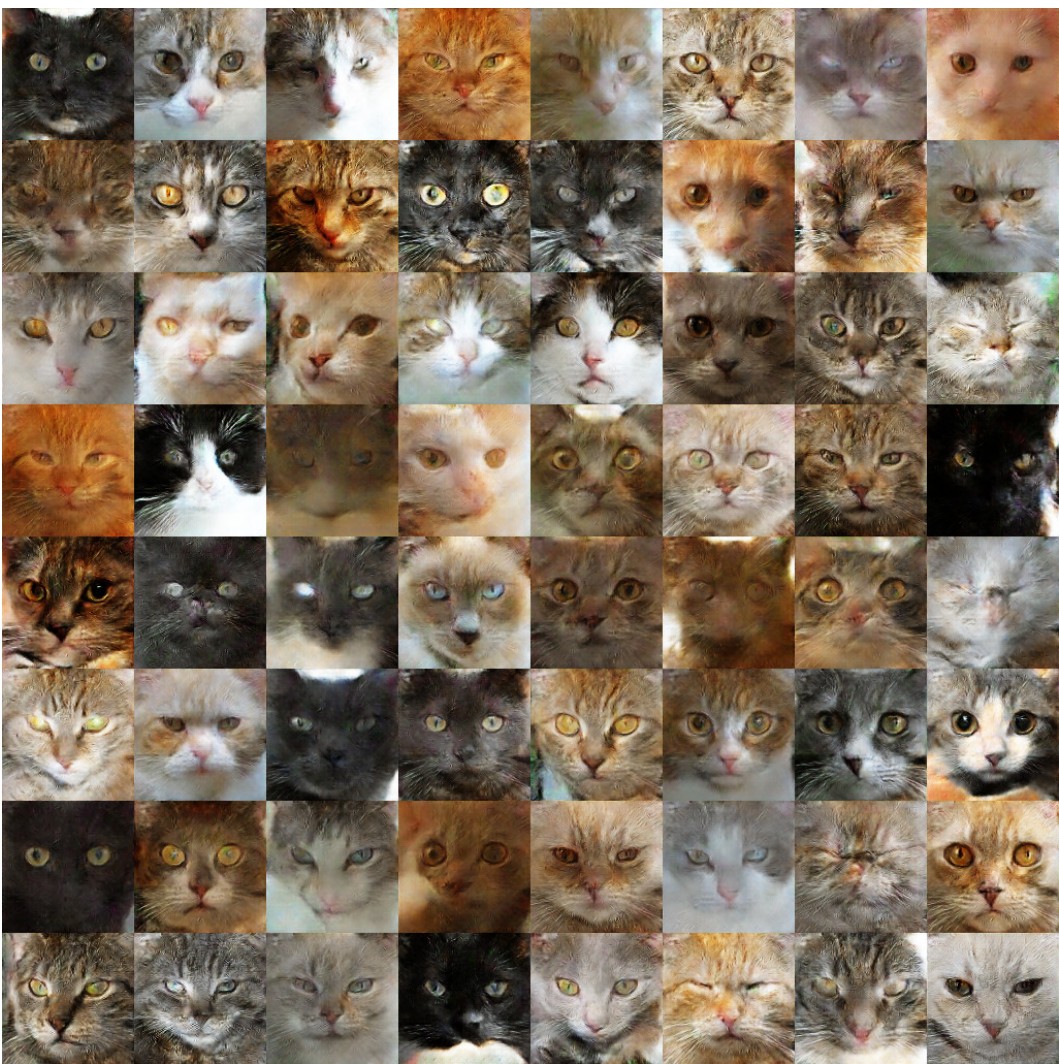

Figure 22b: Early stopping samples from worst MM-nsat run for CAT $128^2$ using DCGAN with spectral normalization in the discriminator and self-attention (as in main text).

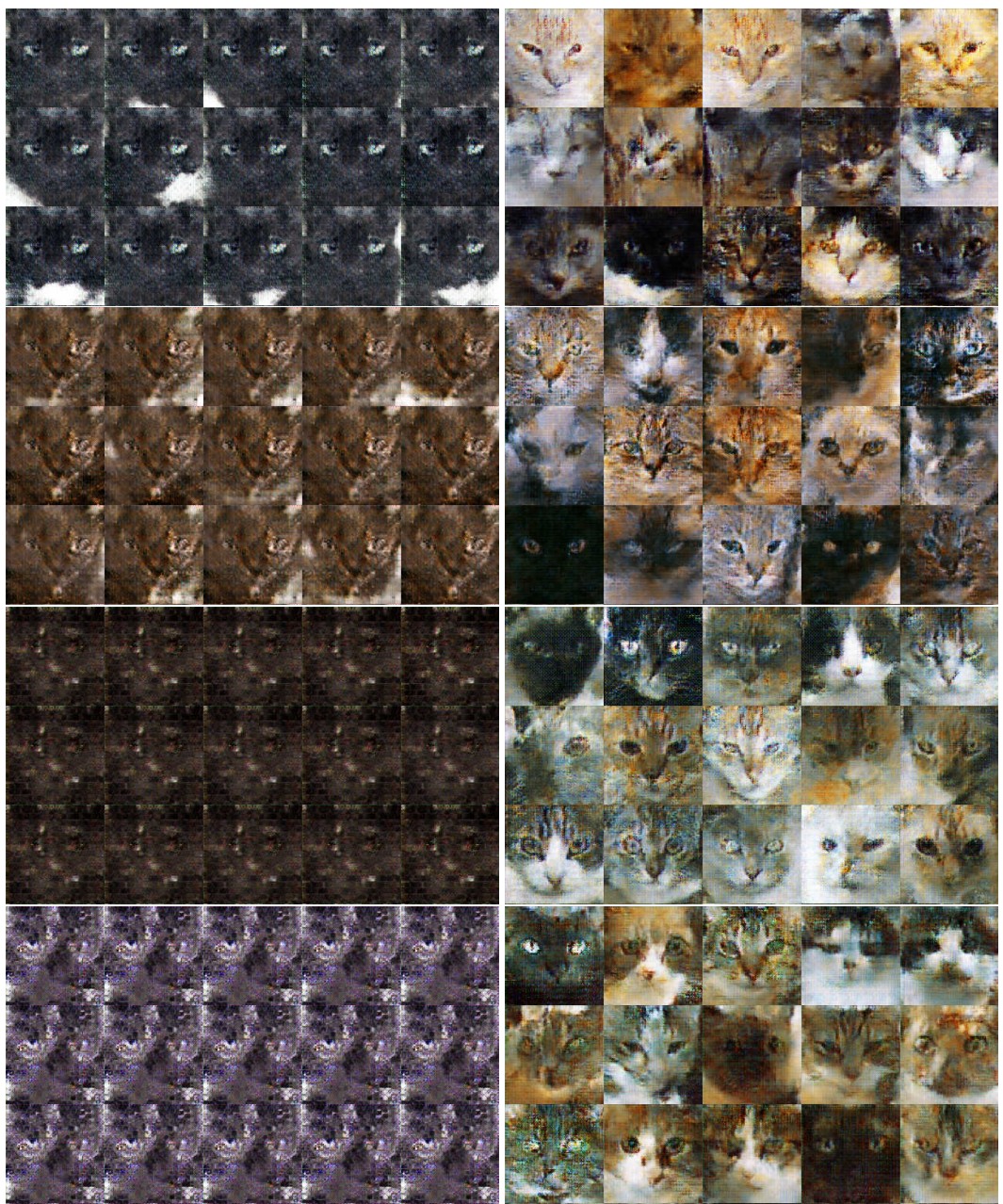

Figure 22c: Final CAT $128^2$ samples for Conv-6 networks for 4 random runs. Left: NS. Right: MM-nsat.

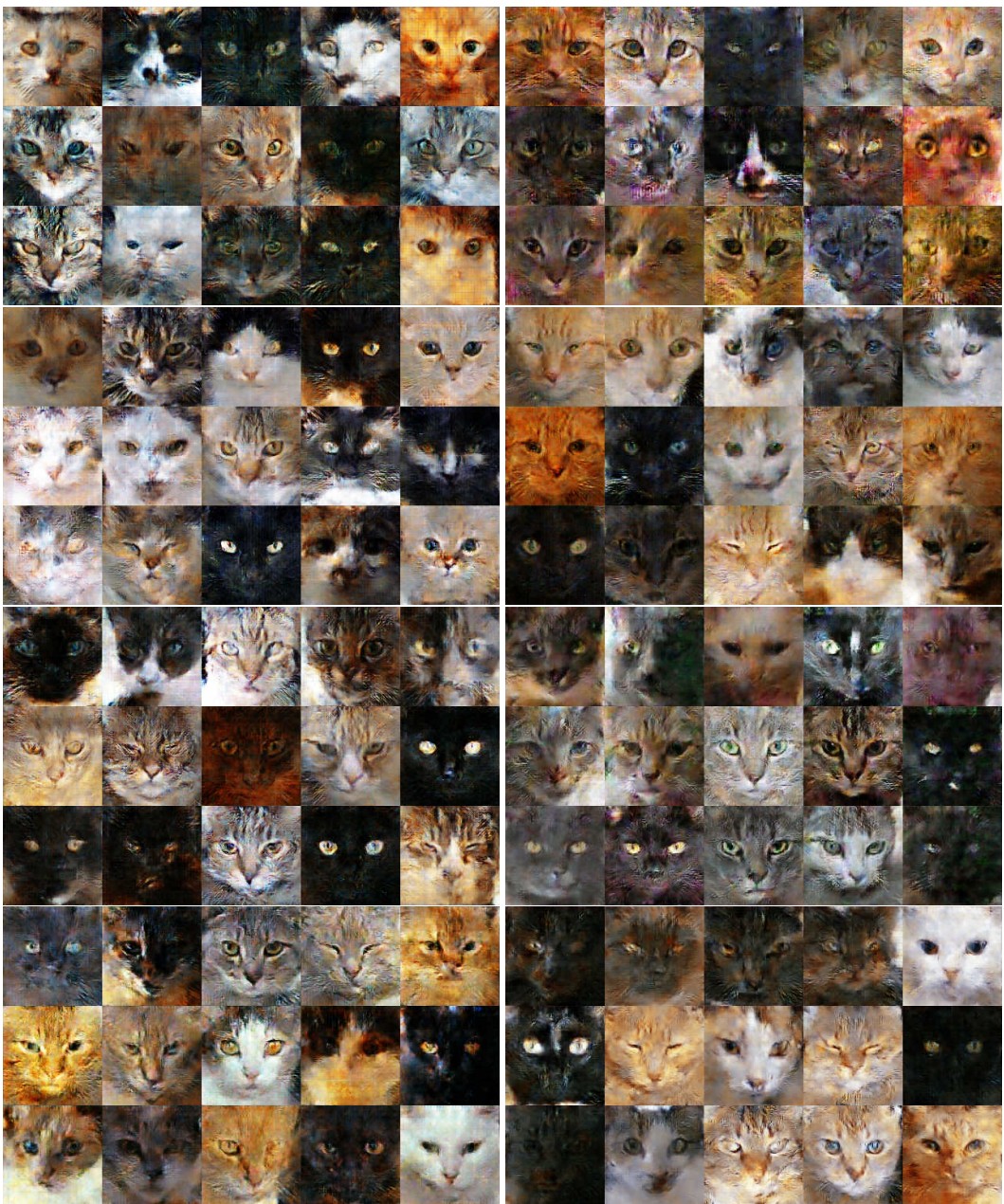

Figure 22d: Final CAT $128^2$ samples for Conv-6-sn networks for 4 random runs. Left: NS. Right: MM-nsat.

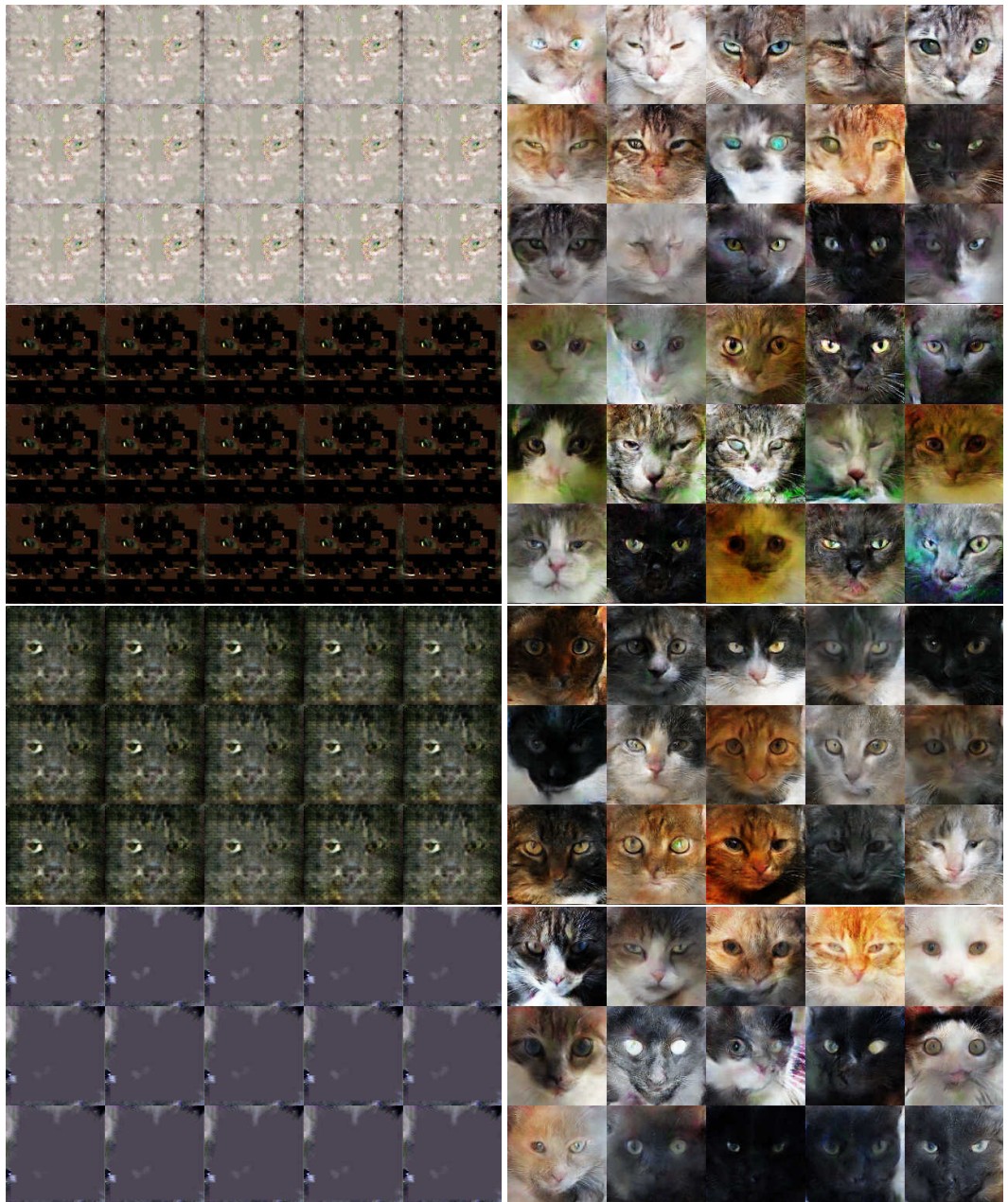

Figure 22e: Final CAT $128^2$ samples for DCGAN-bn-sn networks for 4 random runs. Left: NS. Right: MM-nsat.

# V   FFHQ AT VARIOUS RESOLUTIONS

Figure 23 shows results for training simple convolutional GANs on the FFHQ dataset downsampled to various resolutions, comparing NS-GAN and MM-nsat. The differences between the two cost functions are generally as seen in other experiments. Aside from the decent early stopping results for NS-GAN even at high resolutions the and relatively strong performance of MM-nsat even at $512^2$ resolution with very unsophisticated network architectures, these results only replicate previously discussed effects. Figure 24 shows samples from training at $512^2$, showcasing the mode collapsing behavior that causes FID to increase. Note that samples are JPEG compressed.

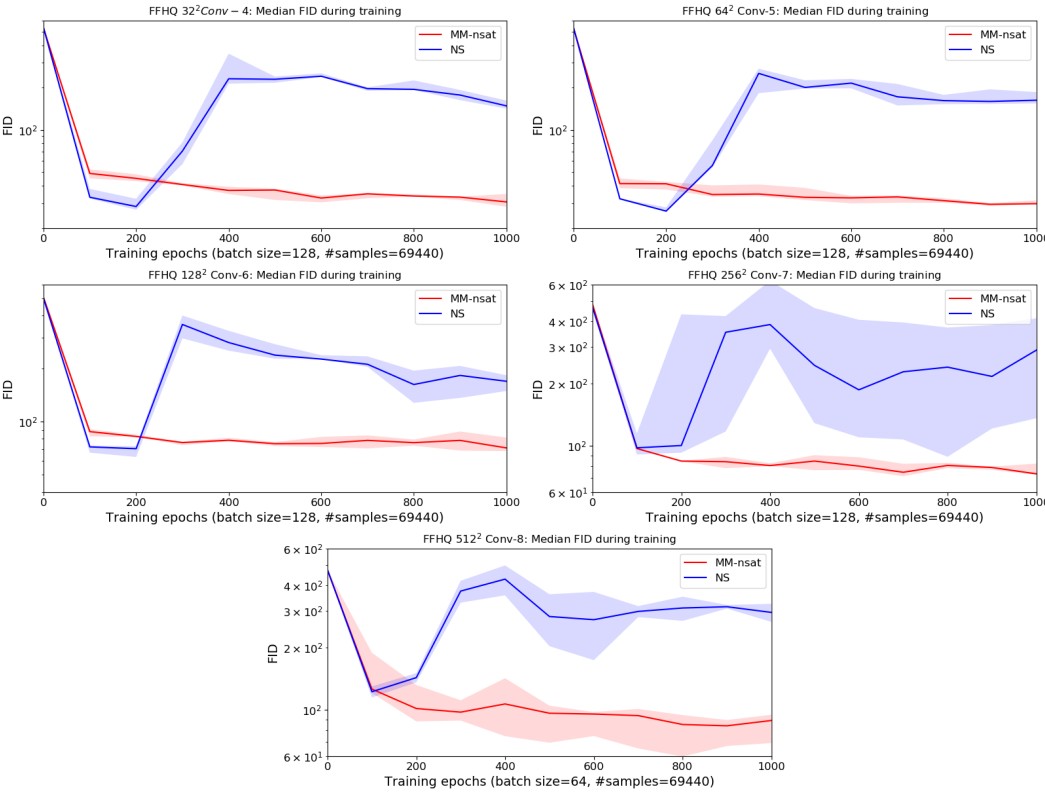

Figure 23: FID, an estimated distance between generated and real data based on Inception activations. For various resolutions using the FFHQ dataset, we plot the median values for three runs during training. The shaded area indicates maximum and minimum values. Results for FFHQ $1024^2$ has been left out: neither MM-nsat or NS-GAN achieve FID values meaningfully better than randomly initialized networks. For resolutions from $32^2$ to $512^2$, training collapse is universal for NS-GAN while MM-nsat is stable and achieves similar FID for low resolutions and much better FID for high resolutions.

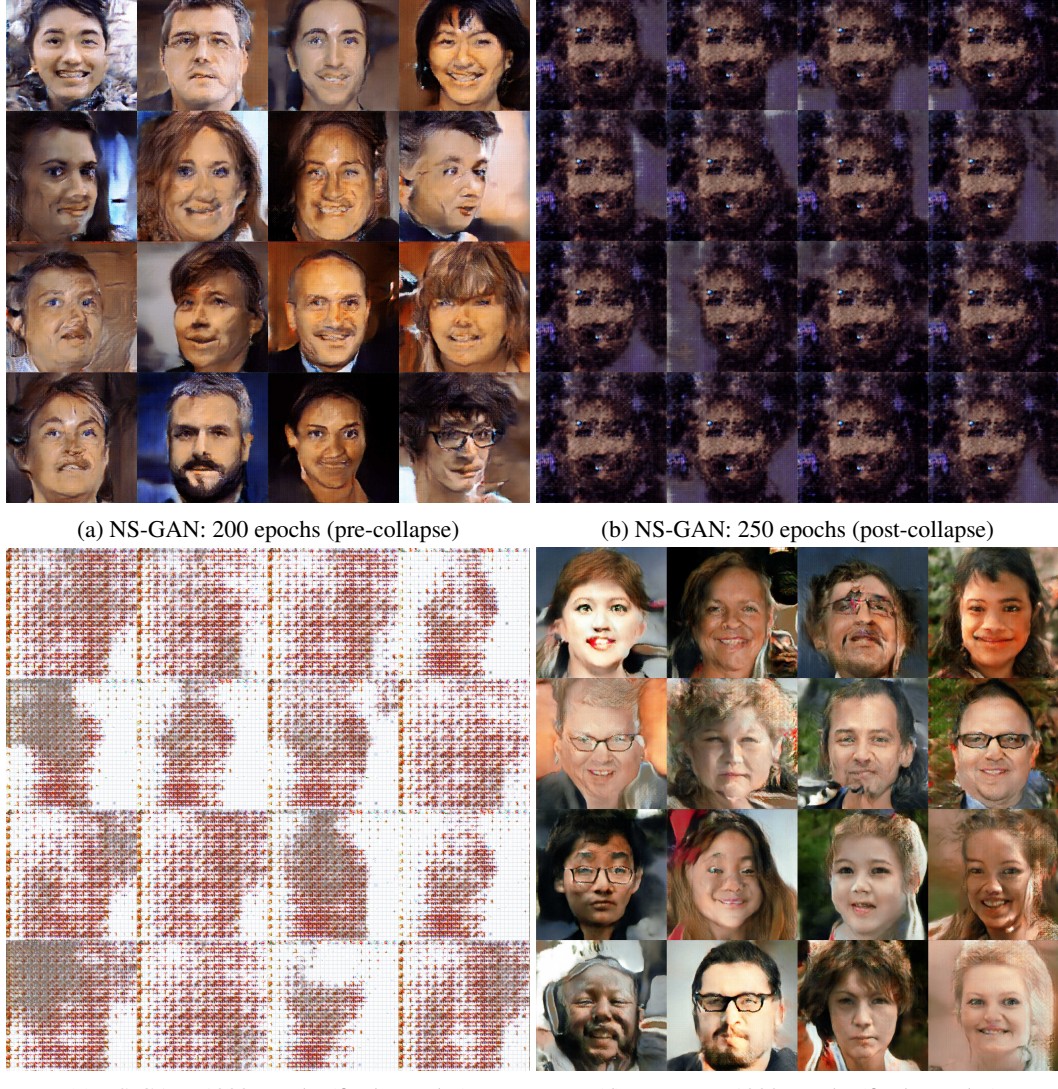

Figure 24: FFHQ $512^2$ samples for 9-layer convolutional networks. Subfigures (a-c) show NS-GAN's early stopping performance, its abrupt, catastrophical mode collapse and its final, nonsensical samples. Subfigure (d) shows final samples for MM-nsat, which fall well short of photo-realism, but are of higher quality than the NS-GAN early stopping samples and fairly good given the exceedingly simple network architecture.

Table 4: FFHQ FID values using StyleGAN, comparing results from retraining the original implementation on 4 GPUs at full and lowered resolution, with results obtained by replacing NS loss with MM-nsat without any additional tuning. Cross-reference with figure 25 showing samples.

| Settings | Final FID | Best FID |
|---|---|---|
| $1024^2$ NS | 4.2887 | 3.9354 |
| $1024^2$ MM-nsat | 6.1292 | 5.9294 |
| $256^2$ NS | 5.6232 | 5.4288 |
| $256^2$ MM-nsat | 6.2375 | 5.3844 |

## W  STYLEGAN

We run experiments training StyleGAN (Karras et al., 2018), replacing the traditional NS loss used in the original implementation with our MM-nsat. Due to available resources, we train each model using only 4 GPUs. We make no other adjustments. We show results in table 4 and figures 25a and 25b.

The primary result from these tests is that MM-nsat performs reasonably well, albeit worse than NS-GAN, as a drop-in replacement in a sophisticated, state of the art implementation. In the full resolution case, which is carefully optimized by the original authors (for instance learning rate adjustments throughout training and a specific value of truncation for the latent input for $G$), NS achieves much better FID and MM-nsat samples seem to have more artifacts. For the lower resolution case, which is a somewhat more fair comparison, FID values are closely matched: NS gets the best final value, whereas MM-nsat has the best early stopping value.

Notably, StyleGAN uses zero centred gradient penalty and minibatch standard deviation features in $D$ (see R). Both of these can serve to limit the failure modes we have shown for NS-GAN. For StyleGAN, coverage is limited, particularly for the latent truncation settings that achieve the best FID (Kynkäänniemi et al., 2019), and visual quality of samples is the first priority.

Implementing MM-nsat for StyleGAN requires some minor modifications, which essentially consist of accumulating values of $D_p$ from parallels in the same way the original implementation does for gradients and rescaling the total gradient before passing it back to the parallels to update parameters for each network clone.

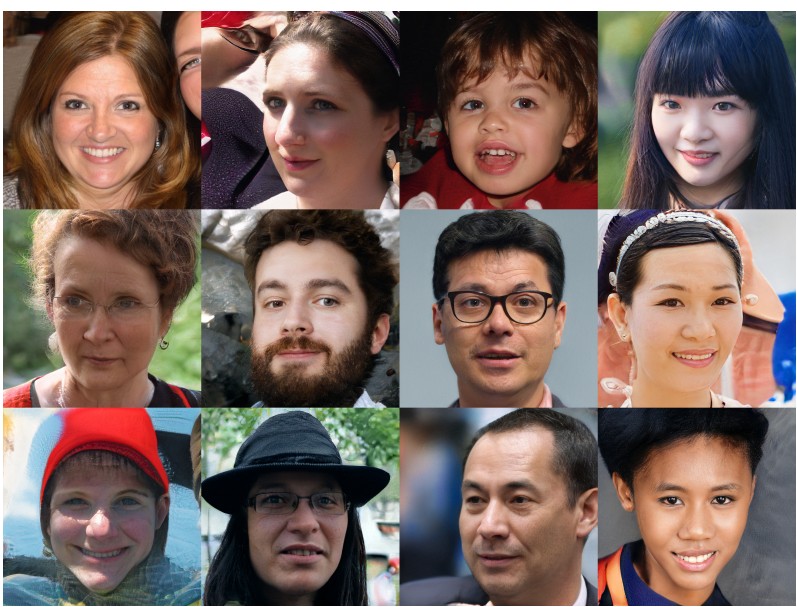

(a) $1024^2$ NS: FID $= 4.2887$
(b) $1024^2$ MM-nsat: FID $= 6.1292$

Figure 25a: FFHQ samples using StyleGAN. Samples selected at random. Overall, MM-nsat samples are somewhat less convincing, especially when viewed in full size.

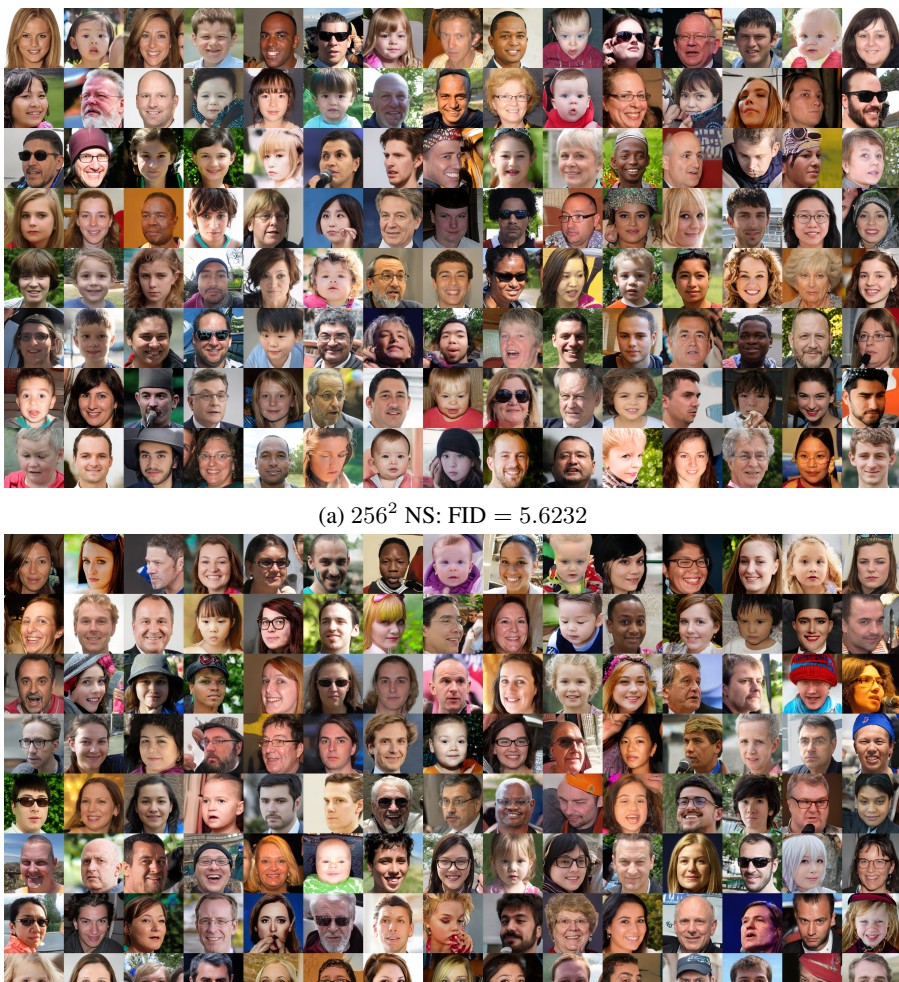

(a) $256^2$ NS: FID = 5.6232

(b) $256^2$ MM-nsat: FID = 6.2375

Figure 25b: Lower resolution FFHQ samples using StyleGAN. Samples selected at random. We find no obvious differences in visual quality.

