# OpenReview forum: "Sample weighting as an explanation for mode collapse in generative adversarial networks"
_ICLR.cc/2021/Conference — Reject_

### Official Review · AnonReviewer1 · 2020-10-21
**An interesting approach, questionable experiments**

**Rating:** 6
**Confidence:** 3

**Review:**

Summary:

This work proposes that many common issues with GAN methods are based on the weighting of the samples given to the generator’s objective function. They focus on a study of the original GAN objective proposed in Goodfellow et al. where the generator’s objective is the negative of the discriminators objective. The GAN community quickly observed that when the discriminator outperforms the generator with this objective, the saturating nature of the sigmoid function causes the gradients to vanish for the generator’s objective. For this reason, a new objective (NS-GAN) was proposed which modifies the generator’s objective to alleviate this gradient vanishing issue.  The authors argue that this modified objective is to blame for a number of common issues with GAN methods -- most notably the mode-dropping issue. The authors present theory that backs up these claims and they propose a new generator training objective which re-weights the gradients of the generator objective to have the same average magnitude as NS-GAN but have the same relative magnitudes of the original GAN objective.

The authors demonstrate the impact of their new loss function in a series of quantitative and qualitative settings.


Strong areas:

I am a very big fan of work that questions standard assumptions that are taken almost as fact within our community. When GANs were originally proposed, most researchers saw the NS-GAN objective as a strict improvement over the MM-GAN objective and moved on. This work does a great job to demonstrate that NS-GAN is most definitely not “superior” to MM-GAN and may possibly be a worse choice of objective function. This change may seem insignificant on the surface and the authors here clearly show that it has an impact -- one large enough to consider if it should be used at all. To aid in their claims, the authors provide clear and concise explanations backed up by easy to understand theory. Particularly interesting to me was the argument that the popular Adam optimizer does not alleviate the saturation issues found with MM-GAN even though my own intuition on the optimizer tells us that it should help with issues like this.

Weaknesses:

The empirical results presented appear promising. The proposed approaches (MM-Unit and MM-NSAT) considerably outperform the NS-GAN objectives. My biggest issues are with the quality of the baselines. I am not an expert in the GAN field, but from inspecting the Conv-4 model with spectral normalization on CIFAR10, it appears like the best performing model achieves an FID of approx 42 and the worst model gets around 48 (figure 8, bottom right). While this difference is notable and the error-bars indicate it is statistically significant, I am concerned because these numbers seem to considerably underperform prior work. The original paper on spectral normalization for GANs reports an FID of 29.3 for standard CNNs. This model uses the NS-GAN objective (to my knowledge) so, I am confused as to why the authors did not simply replicate their setup -- especially since the standard CNN model proposed in the original spectral norm paper can be trained with reasonable compute and the authors have released code. If I am misunderstanding something about the experiments, please do let me know, but I am confused by this choice.

I also took some issues with the D-JS-CD score proposed to score the class distribution of generated samples. There have been a number of proposed metrics like this such as IS and classifier score (https://arxiv.org/abs/1905.10887). I understand that this score (unlike classifier score) does not rely on a conditional model, but if a new score is to be proposed and it is used to convince the reader that a new method performs well, then it should be applied to some baseline models. Pre-trained models from prior GAN papers could be used to obtain these scores.

As a researcher from a different field, my main concern about the experimental results is that the results are quite far from the current state-of-the-field. State of the art results are by no means required for publication in this venue, but the baseline models presented here perform much worse than they have been shown to in prior work. Since a near 20-point increase in FID can be achieved with a few tweaks to the NS-GAN objective (SN-GAN), I am left wondering if the presented improvement from the MM-NSAT objective will vanish once those improvements are applied or if it will still hold. Since this is not shown, then I am uncertain of the significance of the observations made in this work.

Some more nit-picky issues:

The text in the figures is too small and near impossible to read. I would present all of the results in Figure 8 in a table instead. There is no information gained by seeing loss curves. A table would save much more space and allow the readers to more easily compare this work to other works. In Figure 7, I find “best” and “worst” picks by qualitative methods to be somewhat unconvincing. You should show the highest and lowest FID or just show random samples.

My recommendation:

I am not an active member of the GAN community so I am more than willing to accept if my recommendation goes against more senior folks who work in that field.

I found this to be an interesting work that provided a non-trivial insight, backed it up with clear and easy-to-follow theory and demonstrated their observations held on some medium-scale experiments. My biggest issues come from my reservations about the experimental results. The baseline NS-GAN with spectral normalization presented in this work greatly underperforms previously published methods that use the same objective. The difference between the presented baseline and the proposed method is smaller than the difference in performance between the presented baseline and previously published methods with the baseline objective.

These discrepancies give me sufficient doubt where I am not certain that the insights of this work provide a sufficient improvement when combined with architectural improvements and proper parameter tuning.

For these reasons, I am advocating against acceptance of this work but making it clear that I think this paper is borderline.

If the experimental setup was more in line with prior work and the same trend in results held, then I would be more likely to recommend acceptance of this paper.

---

> ### Author Response · Authors · 2020-11-14
> **FID: our results vs Miyato spectral normalization**
>
> Unfortunately, GAN papers tend to use subtly different implementations of FID, such that the numbers are difficult to compare across papers. The number you cite for SpecNorm NS-GAN [Miyato et.al. 2018: Spectral Normalization for Generative Adversarial Networks: https://arxiv.org/pdf/1802.05957.pdf ] uses the following implementation: "We computed the Fréchet inception distance between the true distribution and the generated distribution empirically over 10000 and 5000 samples." Which differs from our implementation, following the convention of the paper introducing FID [Heusel et.al. 2017: (...) Two Time-Scale Update Rule (...): https://arxiv.org/pdf/1706.08500.pdf ]: "We computed the (mw,Cw) on all CelebA images, while for computing (m,C) we used 50,000 randomly selected samples.”
>
> Furthermore, from SpecNorm NS-GAN fig 2 you can see considerable variation in FID depending on the set of hyperparameters used. In our work, we explicitly avoid all hyperparameter tuning, generally using the TensorFlow default values and models with relatively few parameters. To support our theoretical claims in section 2, we are primarily interested in the relative performance of NS and MM-nsat across a variety of settings: for this purpose, we are more concerned with drawing fair comparisons between NS and MM-nsat, than with obtaining strong, quantitative results.
>
> We look more carefully into the discrepancy between SpecNorm NS-GAN and our own results and see if we can pinpoint the reason for the differences. Note that spectral normalization reduces the differences between NS and MM-nsat as discussed in appendix O.

---

> > ### Comment · AnonReviewer1 · 2020-11-22
> > **Thanks!**
> >
> > I thank the authors for their thoughtful response and their choice to run more experiments based on my feedback. I think these additional SN-GAN experiments greatly improve the quality of the paper. The original experiments in the paper do demonstrate the benefits of the proposed change to GANs but those results are very far away from acceptable performance in this day and age. Proposing a new regularizer that improves cifar10 accuracy from 80% to 90% does not tell us much since acceptable performance on that dataset is well above 95%.
> >
> > These new results demonstrate to me that the proposed change retains its positive impact when other improvements are also applied. This is exactly what I hoped to see and I think that these new results greatly improve the paper. I would advise putting them in the main body of the paper!
> >
> > Because of this, I am inclined to change my score to recommend acceptance.

---

> ### Author Response · Authors · 2020-11-14
> **Figure 8 and DJSCD**
>
> We believe it is important to present figure 8 as a figure rather than a table, to highlight the different time dynamics of the MM and NS cost functions. We have had the luxury of inspecting the changes in generated samples over hundreds of different training runs, but presenting all of this information in the paper is not feasible, so we must find some other way of showing the consistent tendency of NS-GAN to produce good samples, but with a gradual loss of diversity over time.
>
> In particular, figure 8 shows large differences between early stopping and final results for NS-GAN. To choose the best model, it is the early stopping result that matters the most, while to understand the theoretical concerns we discuss, the full dynamics including training past the optimal stopping point is informative.
>
> It is for this purpose we use the divergence between real and generated class distributions, enabling us to visualize the process described in section 2.3 taking place in practical experiments. We do not intend DJSCD as a stand-alone metric, but as a diagnostic tool to better explain the differences in FID, between NS itself at different stages of training and between NS and MM-nsat.
>
> It is generally a problem that FID combines sample diversity and quality into a single metric. Much more sophisticated approaches have been attempted to pull these apart, for instance in [Kynkäänniemi et.al. 2019: Improved Precision and Recall (...): https://arxiv.org/abs/1904.06991 ], a method which was criticized in [Naeem et.al. 2020: Reliable Fidelity and Diversity (...): https://arxiv.org/abs/2002.09797 ]. Lacking a solid consensus on how to best approach this problem, we choose to make use of DJSCD as a simple way of visualizing the most critical points.

---

> ### Author Response · Authors · 2020-11-14
> **Practical benefits of MM-nsat**
>
> We share your concerns regarding the ultimate benefits of using MM-nsat: we discuss some of these problems in the text. Our empirical results, while interesting, are primarily meant to support and illustrate our theoretical contributions, and we believe that trying to push state of the art results with MM-nsat would detract from the most important point.
>
> We would still like to highlight our ability to achieve stable training on high-dimensional natural images: see in particular appendices S and T. For a long time, it was received wisdom that training GANs on these kinds of datasets required sophisticated architectures and specialized stabilization techniques: see for instance [Karras et.al. 2017: Progressive growing of GANs (...): https://arxiv.org/pdf/1710.10196.pdf ]. Aside from the Adam optimizer, we use the simplest possible sort of CNNs as in the first GAN paper [Goodfellow et.al. 2014: Generative Adversarial Networks: https://arxiv.org/abs/1406.2661 ], without any of the tricks of stabilization methods introduced since.

---

> ### Author Response · Authors · 2020-11-18
> **First revision**
>
> We have added Supplementary V, which we hope satisfies your concerns regarding comparisons between our results and SpecNorm NS-GAN. Ideally, we would like to expand on the results in this section and integrate it more properly with the rest of the text. For now, it is added at the very end to avoid renaming all subsequent appendices.
>
> Regarding figure 7 (best vs worst results), we are not clear on whether you are aware of the additional results in the supplementary material. We found this to be the most illustrative way to present the results, leaving the more rigorous comparisons for Supplementary S (and T), but are open to making changes. We are similarly very interested to hear whether our explanations regarding figure 8, DJSCS etc are satisfying or if there are improvements we can make.

---

### Official Review · AnonReviewer3 · 2020-10-29
**Bayesians hate this one weird trick for improving GAN training!**

**Rating:** 6
**Confidence:** 4

**Review:**

The authors present a simple estimator which approximates the gradient of the "true" generator loss in GANs using the gradients from the more commonly used non-saturating generator loss. They present results on toy data and small image datasets like CIFAR to show that the method leads to stabler training and does not suffer from mode collapse as badly.

Had this paper appeared at ICLR in, say, 2016 or 2017, I think it would have been a welcome addition to the literature on "tricks to improve GAN training", which was fairly small at that point. Since then, the field has absolutely exploded, to a degree that I find somewhat baffling, as GANs have no practical applications beyond generating pretty pictures. In addition, recent advances to VAE training (building on some of these GAN tricks like spectral normalization) have massively improved the quality of VAE samples, so the need for GANs continues to diminish. Nevertheless, it is not my job to judge whether a given paper is trendy or not - only to judge whether it is technically correct and up to the standards of publication. On that front, I think it is a decent paper - the method is quite simple, but the results on both mixtures of Gaussians and CIFAR and other image datasets do seem like an improvement. I would have appreciated experiments on a larger dataset like CIFAR or high-res CelebA. I also would have appreciated a more rigorous comparison against the plethora of GAN training tricks that have appeared in the last 5 years - for instance, competitive gradient descent (Schaefer and Anandkumar, 2019) to name just one. As it is, I worry this paper will disappear in a flood of other similar papers without more rigorous comparisons - but then, that may be for the field to judge after it is published.

A few stylistic points:
* Please increase the size of the axis labels in Fig 1. They are unreadable unless zoomed in very far on a screen.

---

> ### Author Response · Authors · 2020-11-14
> **Larger datasets and GAN vs VAE**
>
> For higher dimensional results, please note appendices S, T and U. While we do not use CelebA-HQ, we show results for CAT 128x128 and Flicker Faces-HQ at various resolutions.
>
> We follow the competition between GANs, VAEs and other generative models with great interest and are pleased to see the recent, strong results for VAEs. Our contribution is primarily to the theoretical understanding of GANs and we believe its importance is mostly independent of the relative merits of GANs and VAEs. We have taken the liberty of elaborating a bit on this point:
>
> There are two fundamentally different ways of reading our paper: either as theory leading to a training trick that seems to improve performance; or as a theoretical investigation of the importance of the relative weighting of samples which shows a fundamental liability with the widely used NS-GAN formulation, with empirics as supporting evidence. We find this second reading the most interesting.
>
> In particular, we believe that one of the fundamental problems with GANs is that the generator lacks a real data loss term, unlike for instance VAEs, which prevents G from directly observing the increased cost from dropped modes. This makes it important that G's cost function is sensitive to mode dropping when it starts happening. We argue that the implicit sample weightings of the commonly used NS and our MM-nsat cost functions is an important step towards explaining and addressing this problem, which is one of the major shortcomings of GANs compared to VAEs.

---

### Official Review · AnonReviewer4 · 2020-11-01
**IW-GAN revisited**

**Rating:** 6
**Confidence:** 4

**Review:**

This paper reexamines the original (MM) and the non-saturating (NS) GAN objective.  The authors show that the gradients of the respective objectives just differ from a scaling factor depending on the discriminator's output for generated samples. While the scaling factor for the MM gradient is responsible for the well known vanishing gradient if the discriminator is optimal, the scaling factor of the NS gradient counteracts this saturation effect. However, on the other side, the NS scaling factor introduces a mode dropping effect and the inability of the learning dynamci to discover new modes. The authors show additionally that the NS minibatch gradient is the weighted sum of the single sample MM gradients with respective scaling factors as weights. These scaling factors avoid saturating, however, alter the direction of the resulting minibatch gradient. To counteract the change of the gradient direction the authors propose to summarize the sample scaling factors into one scalar for the batch gradient which preserves the non-saturating behaviour of the NS and the gradient direction of the MM objective. The new GAN objective is called MM-nsat.  Additionally the authors discuss the non-saturating effect of the ADAM beta2 parameter for the MM-GAN Generator.

Experiments: A ring of Gaussians experiment shows the mode dropping effect of the NS-GAN. Training a 4-layer network on MNIST shows demonstrates the vanishing gradient effect on MM-GAN and the counteracting effect with a smaller ADAM beta2 parameter applied to the Generator. Also on MNIST MM-nsat was compared with NS showing a lower FID and Jensen-Shannon divergence between real and generated class distributions for MM-nsat. On the Cat 128x128 dataset mode collapse was visually shown for NS.
On MNIST, CIFAR10 and Cat 128x128 MM-nsat was compared with NS, Hinge and LS-GAN outperforming all of them based on the FID.

Pros: The theoretical insight why NS-GAN suffers from mode collapse is novel and interesting. The experiments are convincing and extensive.

Cons: The proposed objective is not novel [1][2], however, it is derived directly from the original MM-GAN objective and is better theoretically motivated.

For completeness, it would be interesting to see the experiments in section K with the pair MM-nsat/MM as well.

[1] R Devon Hjelm, Athul Paul Jacob, Tong Che, Adam Trischler, Kyunghyun Cho, and Yoshua Bengio. Boundaryseeking generative adversarial networks, 2018
[2] Zhiting Hu, Zichao Yang, Ruslan Salakhutdinov, and Eric P. Xing. On unifying deep generative models. CoRR,
abs/1706.00550, 2017. URL http://arxiv.org/abs/1706.00550

---

> ### Author Response · Authors · 2020-11-14
> **MMvs MM-nsat for appendix K**
>
> We agree that MM vs MM-nsat plots in Appendix K would be a good inclusion. Since MM and MM-nsat gradients are parallel by construction, the cosine similarity should always be 1, but the relative gradient magnitudes (or equivalently: the size of the MM-nsat rescaling factor) throughout training is indeed very interesting.
>
> We have this value plotted in messy diagnostic plots not included in the paper. The general trend is that this rescaling factor has a transient spike (roughly 1e2 to 1e8) during a critical, early phase of training (roughly 1 to 10 epochs) and is otherwise around 1e1, slowly increasing towards the end of training. We will comment further on this while adding readable plots in the revised version of the paper.
>
> Thanks also for your very clear explanation of the idea of rescaling for MM-nsat! We will come back later with some additional comments on the wider significance of the connection we show between sample weighting and mode dropping. See for instance appendix E for comments on one recent NeurIPS publication for which our work is highly relevant [Sinha et.al. 2020: Top-K training of GANs (...): https://arxiv.org/abs/2002.06224 ].

---

### Official Review · AnonReviewer2 · 2020-11-10
**Review for Paper**

**Rating:** 6
**Confidence:** 4

**Review:**

This paper proposes an explanation for mode collapse in the original GAN with the log -D objective for the generator (dubbed the non-saturating GAN or NS-GAN for short). The paper takes the approach of comparing the gradient of the generator objective for the original GAN with cross-entropy loss (dubbed the minimax GAN or MM-GAN for short) and the log -D variant. The key observation is that the difference between the gradient of the generator objective of MM-GAN and NS-GAN is that MM-GAN has a factor of D_p(G(z,\theta)), whereas NS-GAN has a factor of 1-D_p(G(z,\theta)), where D_p(G(z,\theta)) is the output of the discriminator on a sample generated from z. Hence, the terms in the MM-GAN gradient that appear fake to the discriminator are downweighted, whereas they are upweighted in the NS-GAN gradient. Because the samples from modes that are already overrepresented are likely declared fake by the discriminator, the contribution to the generator gradient is dominated by these samples in NS-GAN.

Strengths:
This observation is very nice, intuitive and simple and is new to my knowledge. It sheds light on the weaknesses of the log -D variant of GANs.

Weaknesses:
Paper title is broader than what the paper shows - the proposes explanation only applies to GANs with the log -D generator objective and does not apply to other GAN variants, e.g.: original GAN (with cross-entropy generator objective), WGAN, LSGAN etc.

The argument is not presented clearly, and sometimes observations with no obvious logical relationship to the claim are mentioned. At other times, it is unclear what the point is. For example:

On pg. 3, in the first paragraph under sect. 2.3, it is mentioned that "the minibatch used to update G will have more samples from O since they are generated more often". It is unclear how the number of samples from O in the minibatch could cause a difference between MM-GAN and NS-GAN - this observation is true for both MM-GAN and NS-GAN!

In the next paragraph, the paper mentioned that the generator gradient "is only locally informative" - again this is true for all GANs. How is this relevant for the argument made in the paper?

Then it mentioned that "generated samples give rise to conflicting gradients" - it is unclear what conflicting gradients mean. In what sense are the gradients "conflicting"?

In the next paragraph, the paper mentioned "NS-GAN struggles to discover new modes: g(x) ≈ 0 ⇒ 1 − D_p(x) ≈ 0" - the connection between the claim about the difficulty of discovering new modes and the equations should be explained more clearly. Also, the implication in the equations would not be true if r(x) ≈ 0.

Also, for eq. (7), technically the optimal discriminator is only uniquely defined at the real data points (see Sinn & Rawat, AISTATS 2018) because the GAN is trained on a finite sample. The optimal discriminator result in (Goodfellow et al., 2014) assumes access to the true data distribution, or equivalently an infinite stream of samples from the distribution. So technically the paper's claim on the discovery of new modes cannot be justified by "g(x) ≈ 0 ⇒ 1 − D_p(x) ≈ 0", because D_p(x) could take on any value at locations other than the data points. Similarly, on pg. 4, in the paragraph below eq. (9), because D_l^{opt} could take on arbitrary values at positions other than the data points, the claim that D_{opt} = ±\infty is inaccurate - it is in fact not uniquely defined for points that are not real data points. Though because this is a common mistake in the literature, I'd be fine with the addition of a note that explains this caveat before presenting the theoretical argument (without insisting on a fundamental solution to this issue).

Sect. 2.4 on "MM-GAN Interaction with ADAM" is not very mathematically rigorous and relies primarily on an assumption that the value of the logits of the discriminator approaches the optimum linearly. It's unclear if this assumption is actually justified, since the gradient of the cross-entropy loss w.r.t the logits tapers off on the extreme ends of the logits, so the logits should approach the optimum more and more slowly as they become larger in magnitude.

Also, in eq. 11, the left size is vector graphics, whereas the right side is a scalar.

Under sect. 3.1, the JSD between class predictions assume each class contains only one mode and cannot detect intra-class mode collapse. This caveat should be prominently posted.

---

> ### Author Response · Authors · 2020-11-14
> **Title is broader than what the paper shows?**
>
> Edit final revision: added suppplementary Q discussing this matter.
>
> It is true that we mostly focus on the mode dropping effect for NS-GAN. However, the same argument as presented in section 2.3 applies more generally, as suggested by empirical results (fig 11, fig 17). We make a very brief remark at the end of Appendix M touching on the reason for this and will elaborate here:
>
> As you observe, the NS-GAN generator upweights overrepresented generated samples. The combination of upweighting and overrepresentation allows a subset of modes to dominate the parameter update and tends cause other modes to be dropped.
>
> While this combination of effects makes NS-GAN particularly prone to mode dropping, they are not both required. This is most easily observed by considering ½NS + ½MM (fig 12). Since the D_p dependency of the sample weighting cancels out, this linear combination has uniform sample weighting, but its FID still tends to deteriorate during training, much like for Hinge-GAN and LS-GAN.
>
> In other words, overrepresentation by itself is enough to cause mode dropping when using uniform sample weighting. To address gradual mode dropping, samples with low scores should be downweighted. However, this sort of sample weighting would normally mean that gradients decrease when the cost increases, making optimization difficult. For cost functions that downweight overrepresented samples to train properly with normal optimizers, sample weighting and overall gradient magnitude must be decoupled, as done by our minibatch gradient rescaling and also similarly by the importance weight normalization used in [Hu et.al. 2017: On Unifying Deep Generative Models: https://arxiv.org/abs/1706.00550 ].
>
> In other words, commonly used cost formulations do not compensate for overrepresented modes dominating the gradient (LS-GAN, Hinge-GAN, WGAN) and will tend to suffer from a weaker version of effect described in section 2.3 for NS-GAN. Note that our argument only considers cost functions that depend on the samples themselves, and does not necessarily apply to the sort of sample coupling used for instance by relativistic GAN or the interpolation gradient penalty of WGAN-GP.

---

> ### Author Response · Authors · 2020-11-14
> **Unclear exposition in section 2.3**
>
> We apologize for the lack of precision in section 2.3. We have cut too many steps in the argument due to space constraints.  In particular, we mix claims that apply to GANs in general (G’s gradients mostly being sampled from overrepresented modes, gradients only being locally informative) with claims that apply to NS-GAN specifically (overrepresented samples being upweighted, newly discovered modes being downweighted). We will do our best to find a less confusing way to make these points. Similarly, we will expand on the g(x) approx 0 argument.
>
> Hopefully, the following points will be readily understandable from the text itself after we have rewritten the section, but in the meantime we will try to clarify:
>
> Conflicting gradients is a very non-technical formulation – strictly speaking, gradients from samples are conflicting as long as they are non-parallel. In a more intuitive sense, however, optimization based on G(z1) falling in one mode can have different effects on G(z2) falling in another mode:
> - beneficial (reminiscent of transfer learning)
> - neutral (if different latents effectively use different subsets of the network’s parameters)
> - harmful (if it destroys the parameter configuration required to generate the second mode)
> By conflicting gradients, we meant to indicate this last case, but this is not clear in our original submission.
>
> As you remark, it is not special for NS-GAN that sample gradients are only locally informative. However, it is more critical for NS-GAN, because the interaction between overrepresentation and upweighting allows a single mode to dominate the gradient. While MM-nsat also only has locally informative gradients, it pays more heed to gradients from different parts of the data space due to the balancing effect of downweighting overrepresented and upweighting underrepresented samples.

---

> ### Author Response · Authors · 2020-11-14
> **Optimal discriminator**
>
> Edit final revision: added comments and citation start of section 2.3. Thanks for this reference!
>
> This is a very important point. When discussing the optimal discriminator, we mean the function that minimizes J_D specifically for the current configuration of the G (implicitly, its distribution of generated samples). This optimal discriminator is an idealized abstraction, which we should state much more explicitly.
>
> However, we believe that this idealized interpretation of D^opt is well-defined as long as either r(x) or g(x) are nonzero. In section 2.4, we suggest that if generated samples are sufficiently easy to tell apart from real samples, then D is rewarded for making its scores for these fake samples more extreme (D_p → 0 or equivalently D_l → -infty).
>
> We are not sure whether you disagree with our understanding of the optimal discriminator as given by Goodfellow et.al. 2014, or if you think this idealized model is not realistic enough to draw the conclusions we do. We will more carefully into this and would appreciate a clarification if possible.

---

> ### Author Response · Authors · 2020-11-14
> **D_l approaching -infty linearly for Adam model**
>
> Edit final revision: see supplementary L.2, fig 12 and fig 13.
>
> Note also the more in-depth discussion and empirical results in Appendix L.
>
> The linear model for D’s scores is an oversimplification, and your intuition about the loss and the increase in logits eventually tapering off is correct. However, in our plotted diagnostics (these are not yet in the paper), we get D_l(G(z)) on the order of -50 (D_p(G(z)) = 1e-22) before this happens (!).
>
> Furthermore, we do not yet know whether the tapering effect is due to numerical truncation or the loss itself. Note that even though the loss itself is vanishing, the parameterwise partial derivatives for D may still not vanish, due to growth in different parameters multiplicatively reinforcing eachother’s partial derivatives (this is essentially the exploding gradients effect). This can happen because D's optimization problem, assuming it can cleanly distinguish real and generated samples, is very easy.
>
> Regardless, the most important point is that G’s gradient falls beneath the effective renormalization cutoff given Adam’s epsilon hyperparameter much faster than the the increase in D_l(G(z)) tapers off (roughly five epochs vs a hundred epochs). It is very rare for training to recover from this.
>
> We appreciate that our description with words is much less convincing than the forthcoming plots. We will add these and discussion of the behaviors when we revise the text. Thanks for raising this most interesting and incisive point and please let us know if anything remains unclear.

---

> ### Author Response · Authors · 2020-11-14
> **DJSCD intra-class mode collapse**
>
> This is a good point, which we raise in appendix I: “Furthermore, DJSCD is not sensitive to the degree of coverage or collapse within a given mode.” It is important enough that we will happily promote it to the main text. In particular, we will make sure that we no longer mix the terms mode and class as we do here when revising the text.
>
> We will also revise equation 11, thanks for correcting our notation!

---

> ### Author Response · Authors · 2020-11-18
> **First revision**
>
> Helped by your extensive comments, we have made numerous updates: see also [ https://openreview.net/forum?id=oj3bHNSq_2w&noteId=AJsBaQvPpid ].
>
> Given time constraints, we have spent most of our time on experiments and plots for now. We hope the changes we have made satisfy some of your questions: we will focus more on the theoretical issues you raise, such as for the optimal discriminator and the broadness of the title going forward and hope we will be able to make further, substantial improvements during the revision window.

---

### Author Response · Authors · 2020-11-14
**General comments on our follow-up of reviews**

We thank all the reviewers for contributing their time and expertise. We are happy to see that the main idea in our contribution has been well received and that we have gotten so much helpful criticism. We will try to respond to the reviews as follows; feel free to let us know if you expect something different:

 - We have now given preliminary replies to each review. These replies are not comprehensive: we have instead tried to finish this round of replies as early as possible, such that there is time for a running, back-and-forth discussion where useful.
 - Approx Nov 18 we will try to have incorporated the feedback in a revised version of the paper and to provide full response to all comments, including requests for additional results etc.
 - Approaching Nov 24, we will give a finishing statement on on how we have revised our submission and summarize what we consider to be the key strengths and remaining weaknesses of our work.
- Throughout the whole period, we will make an effort to be available and respond quickly to any comments. Please feel free to ask questions or make suggestions.

On posting our response to the reviewers, we realize that they look very long. Please apologize our lack of brevity: we have done our best to answer your criticisms in full.

Best regards, anonymous authors

---

> ### Author Response · Authors · 2020-11-18
> **First revision added**
>
> We have now uploaded a revised version of both our submission and the code. The most important changes:
>  - Rewritten section 2.3: theoretical arguments for NS-GAN mode collapse and use of D^opt (R2)
>  - Added results/discussion supplementary K (fig 10): MM-nsat vs MM comparisons (R4)
>  - Added results/discussion supplementary L (ssec L.2): validity of linear model for Adam saturation (R2)
>  - Added results/discussion supplementary V: replication/comparison Miyato 2018 Spectral Normalization (R1)
>  - Added replication networks / settings (for supplementary V) to attached code
>  - Quality of life improvements for attached code
>
> Other comments:
>  - We have held off on making changes regarding "Title broader than paper shows": we are open both to arguing more explicitly for the broader claim or to make the title more specifically focused on NS-GAN
>  - Since we expect to make further changes, we have not polished formatting such as figure placements
>  - FFHQ plot in figure 1 should include Hinge-GAN and LS-GAN: we have made this change already to show reformatting to make text in figure 1 more readable

---

> ### Author Response · Authors · 2020-11-24
> **Final revision added**
>
> Most important changes:
>  - added Appendix Q: Sample weighting and mode dropping (…)
>  - added citation Sinn & Rawat 2018: Non-parametric estimation (…) (Thanks, R2)
>  - added results Appendix M: CIFAR-10 Spectral Normalization (…): training runs (5→10, as for most other experiments)
>  - improved readability of figures, numerous changes to layout and text
>  - reorganized appendices – note that this reindexes many of them
>
> See also earlier changes: https://openreview.net/forum?id=oj3bHNSq_2w&noteId=AJsBaQvPpid

---

### Author Response · Authors · 2020-11-24
**Closing statement**

Thanks again, to the reviewers and to all involved in organizing this conference. We have done our best to address the points raised by the reviewers, which we have found very helpful for improving our work.

While we hope our submission speaks for itself, we would like to emphasize the theoretical significance of our contribution for the final phase of review: for the very brief introduction of the widely used NS-GAN in the foundational GAN paper[1]; for a different perspective on a recently introduced method to improve training[2]; and, generally, for improved understanding of the perennial issues of training instability and mode dropping.

Best regards, anonymous authors

1: Goodfellow et.al. 2014: Generative Adversarial Networks: https://arxiv.org/abs/1406.2661

2: Sinha et.al. 2020: Top-K training of GANs: Improving GAN Performance by Throwing Away Bad Samples: https://arxiv.org/abs/2002.06224

---

### Decision · Program_Chairs · 2021-01-07
**Final Decision**

**Decision:**

Reject

**Comment:**

I think we did learn something new from this paper, and I think the reviewers all seem to agree with this.
The observation you make about the objective seems correct and interesting (though reviewers and ACs do sometime miss errors), but I have the following complaints that keep me from recommending acceptance:

1. The theory seems right, but in practice, all sorts of GANs with all sorts of objective functions experience "mode collapse",
so it doesn't seem like the issue you point out w/ the NS-GAN objective can be the whole story.
However, we generally don't ask of a paper that it tells the whole story all in one go...

2. I do think the experiments are somewhat poorly done (compared to those for say the median paper about GANs that gets accepted to one of these conferences). Moreover, many people have made similar experimental claims to the ones that are in this paper that haven't held up on more complicated data sets, so I tend to apply more scrutiny to such claims when they're only evaluated on smaller tasks.

3. There have, as R3 points out, been a huge number of papers proposing tricks for training GANs, and some of them work really well.
What I'm missing from this paper is an exploration of the relationship between your observation and those (mostly ad-hoc) tricks.
Does your observation explain why those tricks are necessary?
Does it explain why some existing trick works as well as it does?
If your observation is totally orthogonal to existing tricks, can you get much better performance on a challenging data set by using it?
I don't feel like I got satisfactory answers to those questions.

All this being said, the paper was borderline, and I think if you dealt with some of the complaints above you would have a pretty good shot of getting a revised version accepted at another major machine learning conference.